# STING signalling is terminated through ESCRT-dependent microautophagy of vesicles originating from recycling endosomes

**A list of authors and their affiliations appears at the end of the paper**

Stimulator of interferon genes (STING) is essential for the type I interferon response against a variety of DNA pathogens. Upon emergence of cytosolic DNA, STING translocates from the endoplasmic reticulum to the Golgi where STING activates the downstream kinase TBK1, then to lysosome through recycling endosomes (REs) for its degradation. Although the molecular machinery of STING activation is extensively studied and defined, the one underlying STING degradation and inactivation has not yet been fully elucidated. Here we show that STING is degraded by the endosomal sorting complexes required for transport (ESCRT)-driven microautophagy. Airyscan super-resolution microscopy and correlative light/electron microscopy suggest that STING-positive vesicles of an RE origin are directly encapsulated into Lamp1-positive compartments. Screening of mammalian *Vps* genes, the yeast homologues of which regulate Golgi-to-vacuole transport, shows that ESCRT proteins are essential for the STING encapsulation into Lamp1-positive compartments. Knockdown of Tsg101 and Vps4, components of ESCRT, results in the accumulation of STING vesicles in the cytosol, leading to the sustained type I interferon response. Knockdown of Tsg101 in human primary T cells leads to an increase the expression of interferon-stimulated genes. STING undergoes K63-linked ubiquitination at lysine 288 during its transit through the Golgi/REs, and this ubiquitination is required for STING degradation. Our results reveal a molecular mechanism that prevents hyperactivation of innate immune signalling, which operates at REs.

Stimulator of interferon genes (STING) is an endoplasmic reticulum (ER)-localized transmembrane protein essential for control of infections of DNA viruses and tumour immune surveillance[1]. After binding to cyclic GMP–AMP[2], which is generated by cyclic GMP–AMP synthase[3] in the presence of cytosolic DNA, STING translocates to the Golgi where STING recruits TBK1 from the cytosol[4] and triggers the type I interferon and pro-inflammatory responses through the activation of interferon regulatory factor 3 (IRF3) and nuclear factor-kappa B[5–10].

STING further translocates to lysosomes for its degradation through recycling endosomes (REs), so that the STING-triggered immune signalling is terminated[8,11–16]. The mechanism of how STING, a transmembrane protein on the exocytic membrane traffic, reaches lysosomes has been poorly understood.

Lysosomes are membrane-bound organelles and contain various acid hydrolases to degrade macromolecules including proteins, lipids and nucleotides[17,18]. The degradative substrates in the extracellular

✉e-mail: waguri@fmu.ac.jp; tom_taguchi@tohoku.ac.jp

space or at the plasma membrane are delivered to lysosomes by the endocytic pathway, whereas the ones in the cytosol reach lysosomes by a mechanism designated autophagy[19]. There exist at least three distinct types of autophagy, that is, macroautophagy (delivery of cytosolic substrates to lysosomal lumen via autophagosomes)[20], chaperone-mediated autophagy (CMA; translocation of cytosolic substrates to lysosomal lumen directly across the limiting membrane of lysosomes)[21] and microautophagy (direct encapsulation of cytosolic substrates into lysosomal lumen)[22]. Mechanism and biological consequences of macroautophagy and CMA have been extensively investigated, whereas those of microautophagy remain unclear, in particular, in mammalian cells[23]. In this Article, we examine which autophagic pathway(s) regulates STING degradation.

## Results

### Direct encapsulation of STING into Lamp1[+]

To examine how STING is delivered to lysosomes, $Sting^{-/-}$ mouse embryonic fibroblasts (MEFs) were stably transduced with mRuby3-tagged mouse STING and enhanced green fluorescent protein (EGFP)-tagged Lamp1 (a lysosomal protein), and imaged with Airyscan super-resolution microscopy. Without stimulation, mRuby3-STING localized to a reticular network throughout the cytoplasm (Fig. 1a), suggesting that STING localized at ER[5]. mRuby3-STING diminished 12 h after stimulation with DMXAA (a membrane-permeable STING agonist). In contrast, addition of lysosomal protease inhibitors (E64d/pepstatin A) restored the fluorescence, with mRuby3-STING mostly in the lumen of Lamp1-positive compartments (Lamp1[+]), not at the limiting membrane of Lamp1[+] (Fig. 1b,c and Extended Data Fig. 1a–c). These results suggested that degradation of STING proceeded in lysosomal lumen. The stimulation of STING with double-stranded DNA (herring testis (HT)-DNA) by lipofection[3] also induced STING degradation in lysosomal lumen (Extended Data Fig. 1d,e).

Membrane proteins, such as STING, may have access to lysosomal lumen by three ways, that is, (1) macroautophagy, (2) membrane fusion or (3) direct encapsulation (Fig. 1d). Several reports suggested that STING degradation did not require macroautophagy[12,14,16], and we confirmed this in Atg5 tet-off MEFs in which macroautophagy was impaired in the presence of doxycycline[24] (Extended Data Fig. 1f). Importantly, with lysosomal protease inhibitors, mRuby3-STING accumulated in lysosomal lumen in Atg5-depleted cells (Extended Data Fig. 1g,h). Furthermore, PI3K inhibitors (wortmannin and 3-methyladenine) did not inhibit STING degradation (Extended Data Fig. 2a–e), suggesting that macroautophagy was not involved in the delivery of STING into lysosomal lumen. The other two scenarios can be distinguished by probing the luminal pH of STING vesicles. We exploited an RE protein transferrin receptor (TfnR). TfnR was $C$-terminally tagged with EGFP, which thus faced the lumen of REs. If 'membrane fusion' occurs, the fluorescence of EGFP should be quenched because of its exposure to lysosomal acidic milieu[25]. If 'direct encapsulation' occurs, the

fluorescence of EGFP should linger until two membranes surrounding EGFP are digested by lysosomal lipases. TfnR-EGFP was expressed together with mRuby3-STING. mRuby3-STING started to co-localize with TfnR-EGFP 60 min after DMXAA stimulation (Extended Data Fig. 3a and Supplementary Video 1), showing that STING reached REs by that time[6,8]. Intriguingly, the fluorescence of TfnR-EGFP was detected at lysotracker-positive acidic compartments together with mRuby3-STING 3 h after DMXAA stimulation (Fig. 1e–g and Extended Data Fig. 3b–d). These results suggested that the STING delivery to lysosomal lumen was mediated by 'direct encapsulation'.

We then examined whether lysosomes and/or endosomes encapsulated STING. Cells were stably transduced with mRuby3-STING, mTagBFP2-Rab5 and Lamp1-EGFP, so that endosomes and lysosomes were simultaneously monitored. Three hours after DMXAA stimulation, when STING started to be in acidic compartments (Fig. 1e), STING was found inside Lamp1[+], but not inside Rab5-positive endosomes (Rab5[+]) (Fig. 1h). The quantitation also revealed that at any time point up to 12 h after stimulation, STING was not found inside Rab5[+] (Fig. 1h, i, Extended Data Fig. 4a), EEA1-positive early endosomes (Extended Data Fig. 4b), or LBPA-positive late endosomes (Extended Data Fig. 4b). These results suggested that Lamp1[+] directly encapsulated STING for degradation.

We then performed time-lapse imaging of live cells. Cells were imaged every 0.4 s from 3 h after DMXAA stimulation. We often found that a portion of irregularly shaped mRuby3-STING-positive chunk in close proximity to Lamp1[+] translocated into the lumen of Lamp1[+] (Fig. 2, Extended Data Fig. 4c–e and Supplementary Video 2). During this process, mRuby3-STING appeared not to diffuse along the limiting membrane of Lamp1[+], further supporting the mechanism of 'direct encapsulation'.

### Evidence of 'direct encapsulation' by CLEM

We sought to validate 'direct encapsulation' by another approach. 'Direct encapsulation', but not 'membrane fusion', will result in the generation of a limiting membrane (indicated by an orange arrowhead in Fig. 3f) that surrounds STING vesicles. To examine whether STING vesicles in lysosomal lumen is surrounded by membrane, correlative light and electron microscopy (CLEM) was exploited. Cells were fixed and imaged with Airyscan super-resolution microscopy 3 h after DMXAA stimulation. The same cells were then processed for electron microscopy (EM). Two images in the same region of cells from fluorescence microscopy and EM were aligned according to multiple lysosomal positions (Fig. 3a). The CLEM analysis revealed that a STING-positive chunk inside lysosomes (magenta in Fig. 3b,c and Extended Data Fig. 5a,b) was composed of a cluster of membrane vesicles. Importantly, the cluster of membrane vesicles was surrounded by single membrane (indicated by orange arrowheads), demonstrating that 'direct encapsulation' is a mechanism underlying the STING delivery into lysosomal lumen. The CLEM analysis also showed the nature of STING membranes that were free from or associated with lysosomes (Fig. 3d,e and Extended Data

**Fig. 1 | Direct encapsulation of STING into the lumen of Lamp1-positive compartments. a**, $Sting^{-/-}$ MEFs stably expressing mRuby3-STING and Lamp1-EGFP were treated with DMXAA. For the inhibition of lysosomal proteolysis, E64d and pepstatin A were added to the medium. Cells were fixed and imaged. **b**, The fluorescence intensity of mRuby3-STING in **a** was quantified. **c**, Cells were stimulated with DMXAA in the presence of E64d/pepstatin A for the indicated times. Data are presented as the ratio (%) of [mRuby3-STING in Lamp1-positive areas (Lamp1[+])]/[mRuby3-STING in whole cell]. **d**, (1) 'Macroautophagy'; STING vesicles are first occluded into autophagosomes, which then fuse with lysosome. (2) 'Membrane fusion'; STING vesicles fuse with endosome or lysosome, followed by invagination of limiting membrane of endosome or lysosome, yielding intraluminal STING vesicles. (3) 'Encapsulation by endosome or lysosome'; STING vesicles are directly encapsulated into endosome or lysosomes. **e–g**, TfnR-EGFP and mRuby3-STING were stably expressed in $Sting^{-/-}$ MEFs. Cells were treated with DMXAA and then with LysoTracker Deep Red. The boxed area in the bottom panels is magnified in the top panels (**e**). Fluorescence intensity

profile along the white line in **e** is shown (**f**). Cells were treated with DMXAA or HT-DNA and then with LysoTracker Deep Red. Data are presented as the ratio (%) of [TfnR-EGFP in LysoTracker-positive areas (LysoTracker[+])]/[TfnR-EGFP in whole cell] (**g**). **h**, $Sting^{-/-}$ MEFs stably expressing mRuby3-STING, Lamp1-EGFP and mTagBFP2-Rab5 were treated with DMXAA. The white boxed area is magnified in the right panels. mTagBFP2-Rab5-positive area and Lamp1-EGFP-positive area are magnified at the bottom, respectively. The fluorescence intensity of mRuby3-STING within Rab5[+] or Lamp1[+] compartments was quantified. **i**, EGFP-Rab5 or Lamp1-EGFP was stably expressed in $Sting^{-/-}$ MEFs reconstituted with mRuby3-STING. Data are presented as the ratio (%) of [mRuby3-STING inside Rab5[+] or Lamp1[+]]/[mRuby3-STING in whole cell]. NS, not significant. Scale bars, 5 μm (**a**), 10 μm (**e,h**) and 1 μm (magnified images in **e** and **h**). Data are presented in box-and-whisker plots with the minimum, maximum, sample median and first versus third quartiles (**b,c,g–i**). The sample size (n) represents the number of cells (**b,c,g,i**) or vesicles (**h**). Source numerical data are available in source data.

Fig. 5c,d). These irregularly shaped STING-positive chunks were indeed clusters of vesicles with electron-dense coat (60–130 nm in diameter) (Fig. 3g). We also performed the CLEM analysis with Rab11, the authentic RE protein, and found that STING within lysosomes co-localized with Rab11 6 h after stimulation (Fig. 3h,i and Extended Data Fig. 6). Together with the data of live-cell imaging (Fig. 1e,h), these results suggested that a cluster of vesicles with an RE origin was directly encapsulated into Lamp1[+].

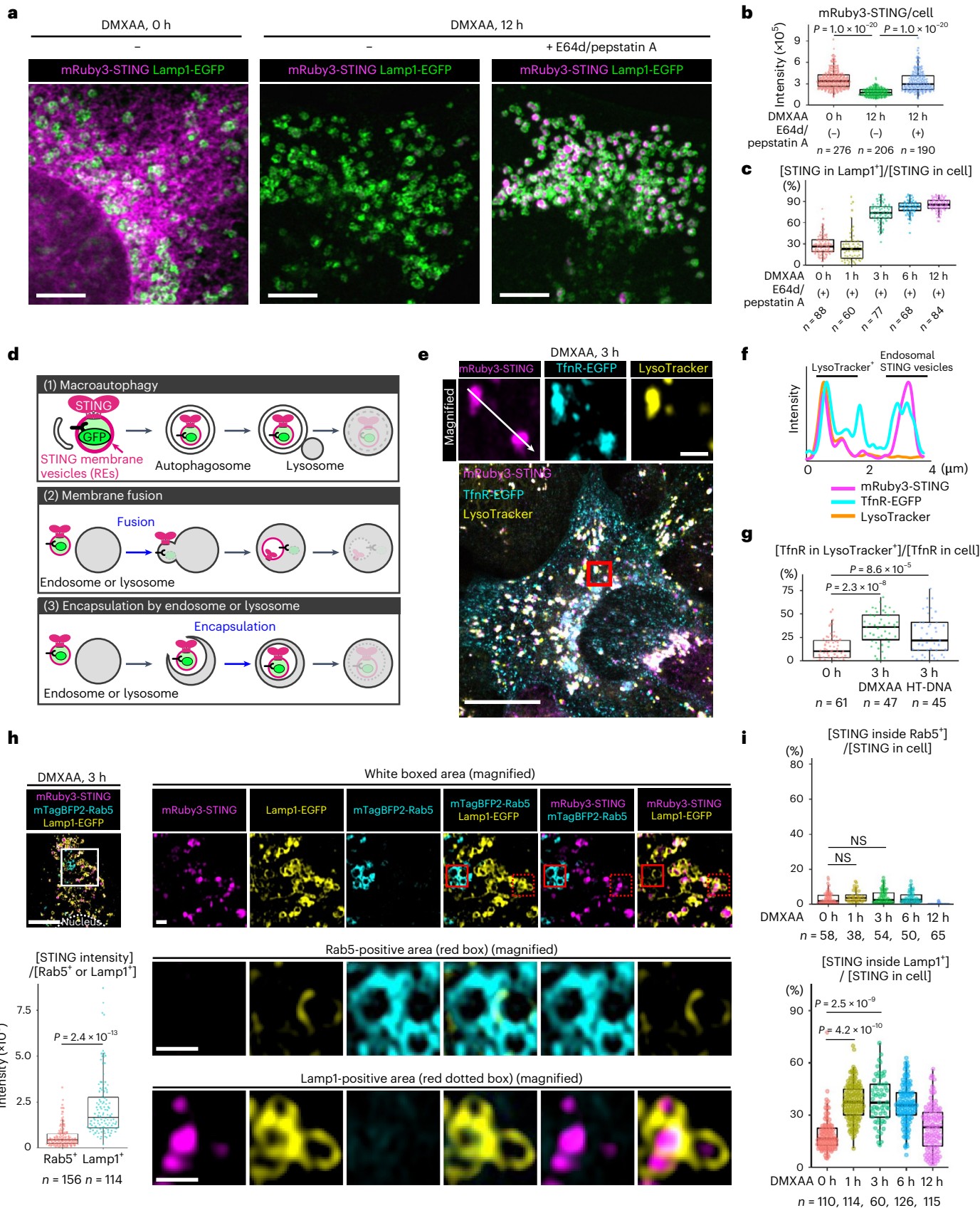

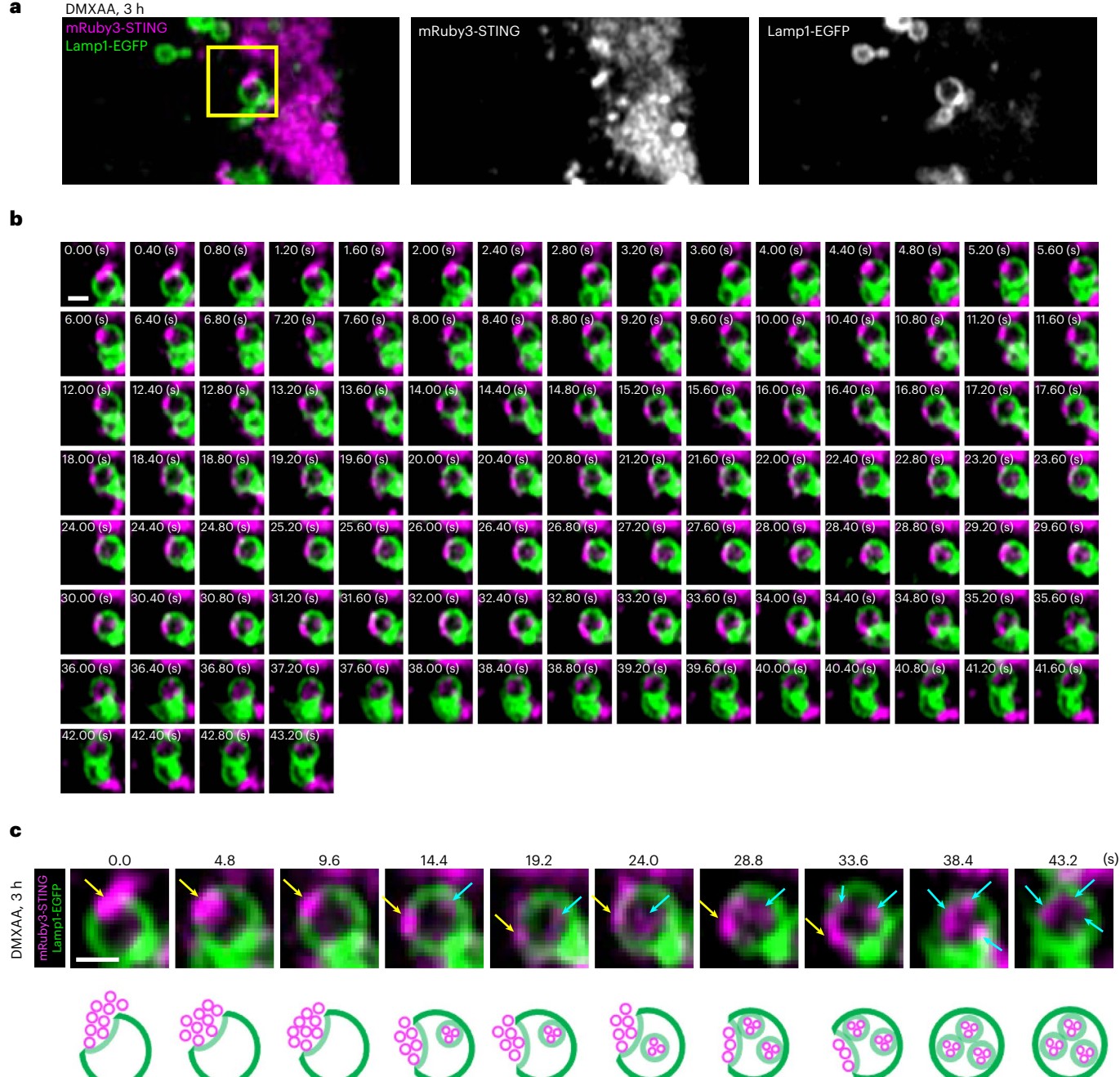

**Fig. 2 | Evidence of 'direct encapsulation' of STING by live-cell imaging.**
**a–c**, *Sting*[−/−] MEFs stably expressing mRuby3-STING and Lamp1-EGFP were imaged by Airyscan super-resolution microscopy every 0.4 s from 3 h after DMXAA stimulation (related to Extended Data Fig. 4b–d): the perinuclear region of cell (**a**); the time-lapse images of the region outlined by the yellow box in **a** shown sequentially (**b**); the schematic corresponding to the individual time-lapse images (**c**). The yellow arrows indicate a cytosolic STING chunk in close proximity to the limiting membrane of Lamp1[+]. A cytosolic STING chunk is depicted as the cluster of vesicles (see also Fig. 3). The cyan arrows indicate STING inside Lamp1[+]. Scale bars, 500 nm (**a–c**).

The observation that STING-positive vesicles had electron-dense coat with 60–130 nm diameter led us to examine if these were clathrin-coated vesicles. EGFP-STING co-localized well with clathrin heavy chain (CHC) 2 h after DMXAA stimulation (Fig. 3j). Knockdown of CHC inhibited the degradation of STING, arresting STING in TfnR-positive compartments (Fig. 3k). These results suggested that packaging of STING into clathrin-coated vesicles at REs was essential process for STING degradation. In line with this notion, the contribution of clathrin-adaptor AP-1 and clathrin in STING degradation has recently been reported in ref. [26].

## ESCRT regulates STING degradation and signalling

In yeast, more than 40 proteins were designated vacuolar protein sorting (Vps) proteins[27,28], which function in sorting of newly synthesized vacuolar proteins from late Golgi to vacuole (the yeast equivalent of lysosome). Given the analogous trafficking pathways that STING and vacuolar proteins follow, we reasoned that mammalian Vps homologues regulate STING traffic to lysosomes. The impaired traffic of STING to lysosomes should result in the suppression of STING degradation[8,11–16], and may also in the duration of the STING-triggered type I interferon response.

We screened 75 *Vps* mammalian homologues with short interfering RNAs (siRNAs) in two criteria, that is, the effect on STING degradation and termination of the type I interferon response. The degree of degradation and that of the type I interferon response were quantitated using flow cytometer and type I interferon bioassay, respectively (Fig. 4a). Knockdown of 55 *Vps* genes showed enhanced suppression of STING degradation, compared with that with control siRNA (Fig. 4b and Extended Data Fig. 7a). Atp6v1b2, a component of subunit B of vacuolar ATPase (v-ATPase), was included in this assay as a positive control. Knockdown of 40 *Vps* genes showed an increased type I interferon response, compared with that with control siRNA (Fig. 4c and Extended Data Fig. 7b). The genes that were ranked within top 25 in both criteria were selected and listed (Fig. 4d). These genes included 4 *Vps* genes (*Vps28*, *Tsg101*, *Vps37a* and *Chmp4b*) that belong to ESCRT[29], *Vps4* (AAA-ATPase) and *Vps39* (a subunit of homotypic fusion and vacuole protein sorting (HOPS) complex). Knockdown of these genes significantly enhanced the expression of Cxcl10, a STING-downstream gene, compared with that with control siRNA (Fig. 4e and Extended Data Fig. 7c), corroborating the results with the type I interferon bioassay (Fig. 4c).

We also performed proteomic analysis of FLAG-STING-binding proteins, aiming at identifying proteins that regulate STING degradation at lysosomes. The amount of individual proteins in the immunoprecipitates by anti-FLAG antibody was quantitated before and after stimulation (Supplementary Table). We selected the proteins, the amount of which increased after stimulation, and further screened them if they were annotated to 'lysosome' in Gene Ontology in Uniprot. This approach led to identify three Vps proteins (Vps4a, Vps4b and Tsg101) (Fig. 4f). Together with the aforementioned results (Fig. 4d), we examined the role of Vps4a, Vps4b and Tsg101 in lysosomal degradation of STING in the subsequent experiments.

### ESCRT functions in encapsulation of STING into Lamp1[+]

We sought to identify the site of actions of Tsg101 and Vps4a/4b, thus examining the trafficking of STING in Tsg101- or Vps4a/b-knockdown cells. In cells treated with control siRNA, the fluorescence of mRuby3-STING diminished entirely 12 h after stimulation with DMXAA, because of its lysosomal degradation (Fig. 5a). In contrast, in cells treated with Tsg101 or Vps4a/b siRNA, the fluorescence of mRuby3-STING lingered and co-localized with TfnR (Fig. 5a,b), indicating that the transport of STING from REs to lysosomes was impaired. Phosphorylated TBK1 (pTBK1), a hallmark of STING activation, also lingered and co-localized with mRuby3-STING in cells treated with Tsg101 or Vps4a/b siRNA (Fig. 5a,b), being consistent with the duration of the STING signalling in these cells (Fig. 4c,e). CLEM analysis of Tsg101 or Vps4a/b siRNA-treated cells showed a cluster of STING-positive vesicles that were peripherally associated with lysosomal limiting membrane (Fig. 5c–f and Extended Data Fig. 8). These results suggested that Tsg101 and Vps4a/4b were essential for encapsulation of a cluster of STING-positive vesicles into lysosomal lumen.

The role of Tsg101 in STING degradation was also confirmed with more physiological stimulations. STING degradation triggered by HT-DNA was significantly retarded in Tsg101-knockdown MRC-5 cells (normal embryonic lung fibroblasts) (Fig. 6a–c). In these cells, phosphorylated STING (pSTING), a hallmark of STING activation, lingered 12 h after stimulation (Fig. 6b,c). Knockdown of Tsg101 in human primary T cells led to an increase of the expression of interferon-stimulated genes, such as IFIT1 and IFI27 (Fig. 6d–f). STING degradation triggered by the infection of herpes simplex virus-1 (HSV-1) was also retarded in Tsg101-knockdown primary MEFs (Fig. 6g). In these cells, pSTING and pTBK1 lingered 8 h after infection (Fig. 6g), and endogenous STING accumulated at Lamp1-negative perinuclear compartments (Fig. 6h,i). Given the expression levels of ICP4, a viral protein produced immediately after infection, Tsg101 knockdown did not interfere with the infection of HSV-1.

### K63 ubiquitination on K288 regulates STING degradation

Given that STING undergoes ubiquitination after stimulation[30] and Tsg101 binds to ubiquitinated proteins[31,32], we reasoned that the binding of Tsg101 to ubiquitinated STING would be required for STING degradation and thus termination of the STING signalling. We confirmed the stimulation-dependent ubiquitination of STING by co-immunoprecipitation analysis. STING became extensively ubiquitinated 2 h after DMXAA stimulation (Fig. 7a).

We sought to examine the dynamics of ubiquitin with STING stimulation. *Sting*[−/−] MEFs were stably transduced with mRuby3-STING and mNeonGreen-ubiquitin and imaged with Airyscan super-resolution microscopy. As with EGFP-STING[8], mRuby3-STING translocated to the perinuclear Golgi by 30 min after DMXAA stimulation, and then to REs by 120 min (Extended Data Fig. 9a). mNeonGreen-ubiquitin distributed diffusively throughout the cytosol and was then translocated to several puncta 120 min after stimulation. These mNeonGreen-ubiquitin puncta were positive with Rab11 (an RE protein) and mRuby3-STING (Extended Data Fig. 9b–d), suggesting that STING at REs was ubiquitinated. Live-cell imaging showed essentially the same results: mNeonGreen-ubiquitin was recruited to mRuby3-STING-positive puncta 95 min after stimulation, at the timing when STING localized at REs (Fig. 7b,c and Supplementary Video 3).

We focused on six conserved lysine residues (K19, K150, K151, K235, K288 and K337) between human and mouse, and generated STING mutants with lysine-to-arginine substitutions individually. Among them, K288R mutant entirely lost the stimulation-dependent ubiquitination (Fig. 7d), and most importantly, was resistant to degradation (Fig. 7d,e), in line with the previous report with HEK293T cells[33]. K288R mutant, as wild-type (WT) STING, translocated from the ER to the Golgi and eventually to REs upon stimulation (Fig. 7f). In cells expressing K288R, the signals of pTBK1 and pIRF3 lingered (Fig. 7g) and the transcription of Cxcl10 was enhanced (Fig. 7h). The immunofluorescence and biochemical analyses showed that STING was subjected to K63-linked ubiquitination on K288 4 h after stimulation (Fig. 7i–k). Thus, these results demonstrated that K63-linked ubiquitination on K288 was required for STING degradation and termination of STING signalling.

---

**Fig. 3 | Evidence of 'direct encapsulation' of STING by CLEM. a–g**, *Sting*[−/−] MEFs stably expressing mRuby3-STING (magenta) and Lamp1-EGFP (green) were treated with DMXAA in the presence of E64d/pepstatin A/orlistat (lipase inhibitor): Lamp1-positive endosomes/lysosomes and STING-positive vesicles (or structures) were identified by Airyscan super-resolution microscopy before processing for transmission EM to examine their ultrastructure (**a**); magnification of the boxed areas in **a** (**b–e**), with orange arrowheads in **b** and **c** indicating the membrane that surrounds STING vesicles (for EM images of serial sections, see Extended Data Fig. 5); a graphical image of lysosome containing membrane-encapsulated STING vesicles (**f**) (green arrowheads indicate limiting membrane of lysosome); the diameter of STING-positive membrane vesicles was measured and plotted as histograms (**g**). **h**, *Sting*[−/−] MEFs stably expressing mRuby3-STING (magenta), Lamp1-EGFP (cyan) and Halo-Rab11a (green) were treated with DMXAA for 6 h in the presence of E64d/pepstatin A/orlistat. **i**, The fluorescence intensity of Halo-Rab11a in lysosomes (Lamp1-positive areas) or in whole cell was quantified. Data are presented in box-and-whisker plots with the minimum, maximum, sample median and first versus third quartiles as the ratio (%) of [Halo-Rab11a in Lamp1[+]]/[Halo-Rab11a in whole cell]. **j**, *Sting*[−/−] MEFs stably expressing EGFP-STING (green) were treated with or without DMXAA. Cells were immunostained with anti-clathrin-heavy chain (CHC) antibody (magenta). **k**, TfnR-EGFP (green) and mRuby3-STING (magenta) were stably expressed in *Sting*[−/−] MEFs. Cells were treated with the indicated siRNAs, and then stimulated with DMXAA. Scale bars, 10 μm (**a**,**j**,**k**), 500 nm in (**b–e**,**h**) and 500 nm (magnified images in **j** and **k**). The sample size (*n*) represents the number of cells (**i**) or vesicles (**g**). Source numerical data are available in source data.

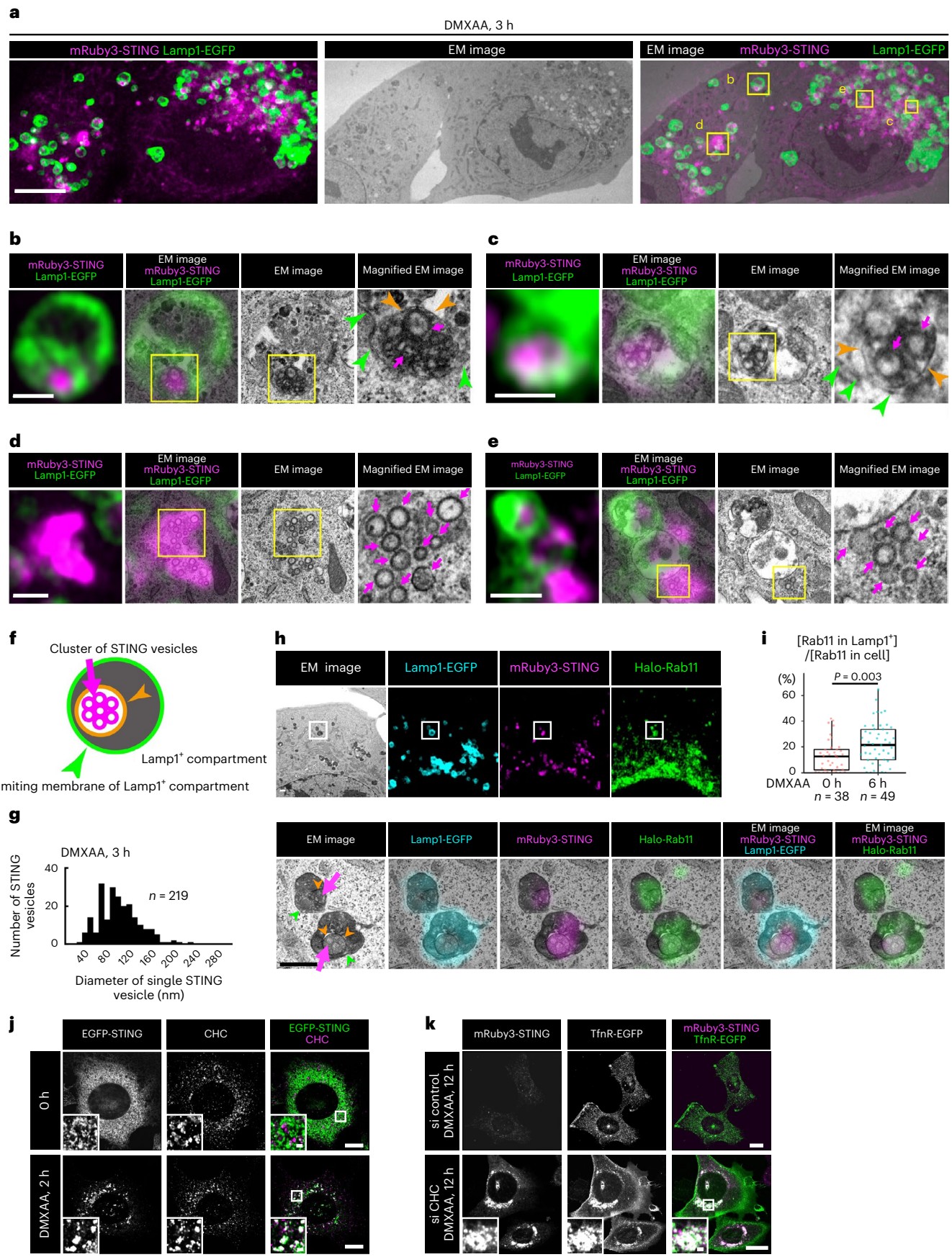

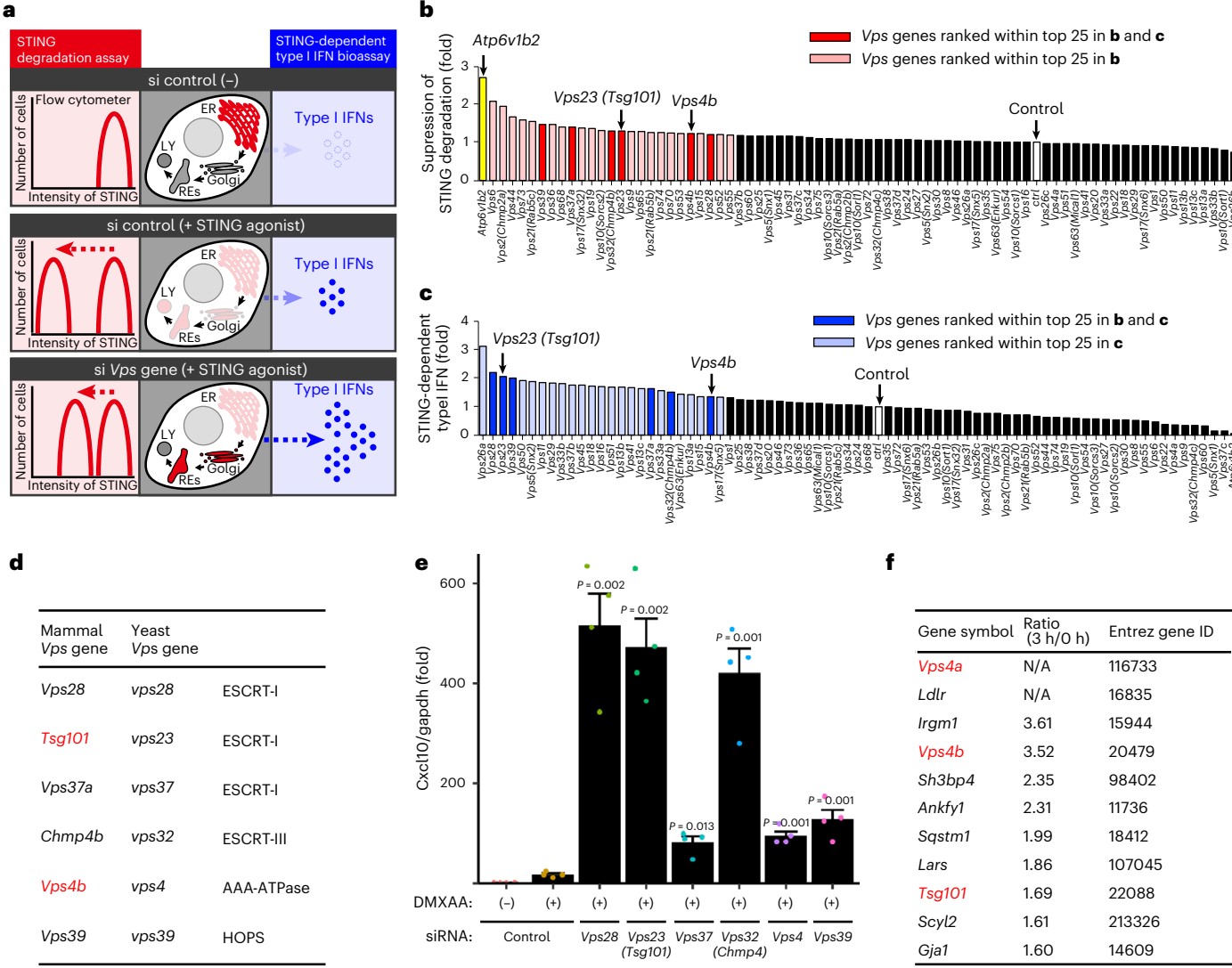

**Fig. 4 | Mammalian *Vps* genes essential for STING degradation and termination of type I interferon response. a**, Schematic overview of the screening procedures. **b**, Screening of mammalian *Vps* genes required for STING degradation. *Sting*[−/−] MEFs reconstituted with mRuby3-STING were treated with siRNA against individual *Vps* genes, and stimulated with DMXAA for 18 h. Cells were analysed by flow cytometry. MFI of mRuby3 in stimulated cells was divided by MFI of mRuby3 in the corresponding unstimulated cells. The calculated value from cells treated with *Vps* siRNA was then normalized to that of cells treated with control siRNA. The top 25 genes are highlighted in red. Bright red bars indicate the genes that were also ranked within top 25 in **c**. **c**, Screening for mammalian *Vps* genes required for suppression of STING-dependent type I interferon response. MEFs were treated with siRNA against individual *Vps* genes, and stimulated with DMXAA for 10 h. Cell supernatants were analysed for type I interferon (IFN). IFN activity from cells treated with *Vps* siRNA was normalized to that of cells treated with control siRNA. The top 25 genes are highlighted in blue. Bright blue bars indicate the genes that were also ranked within top 25 in **b**.

**d**, *Vps* genes ranked within top 25 both in **b** and **c** are shown. **e**, The expression of Cxcl10 in MEFs that were treated with siRNA against the indicated *Vps* genes, and then stimulated with DMXAA for 12 h. Data are presented as mean ± standard deviation (s.d.). Statistical significances between control siRNA/DMXAA (+) and the indicated siRNAs/DMXAA (+) were determined by performing Student's unpaired *t*-test with Bonferroni multiple correction. **f**, FLAG-STING-reconstituted *Sting*[−/−] MEFs were stimulated with DMXAA for 3 h, and lysed. FLAG-STING in the lysates was immunoprecipitated. Co-immunoprecipitated proteins were identified by MS. The ratio of abundance of identified proteins before and after stimulation was then calculated individually. The listed are lysosomal proteins that showed increased abundance after stimulation. Gene Ontology analysis in Uniprot was performed to identify lysosomal proteins. N/A indicates a protein that was not detected without stimulation. The sample size (*n*) represents the number of the biological replicates (**e**). Source numerical data are available in source data.

## UEV domain of Tsg101 is essential for STING degradation

We next examined whether Tsg101, a ubiquitin-binding protein, was required for the degradation of ubiquitinated STING. The smeared bands corresponding to ubiquitinated EGFP-STING diminished 12 h after stimulation in control cells, but not in cells treated with Tsg101 siRNA (Fig. 8a). In accordance with these biochemical data, the fluorescence signals of mRuby3-STING and K63-linked ubiquitin diminished 12 h after stimulation in control cells, but not in cells treated with Tsg101

siRNA (Fig. 8b–e). Of note, mRuby3-STING or K63-linked ubiquitin accumulated outside Lamp1[+] in Tsg101-depleted cells (Fig. 8b–e and Extended Data Fig. 9e,f). These results suggested a role of Tsg101 in encapsulation of K63-linked ubiquitinated STING into lysosomes.

Finally, we examined a role of an *N*-terminal ubiquitin E2 variant (UEV) domain of Tsg101 in the termination of the STING signalling. EGFP-Tsg101 distributed diffusively throughout the cytosol before stimulation and was translocated to several mRuby3-STING-positive

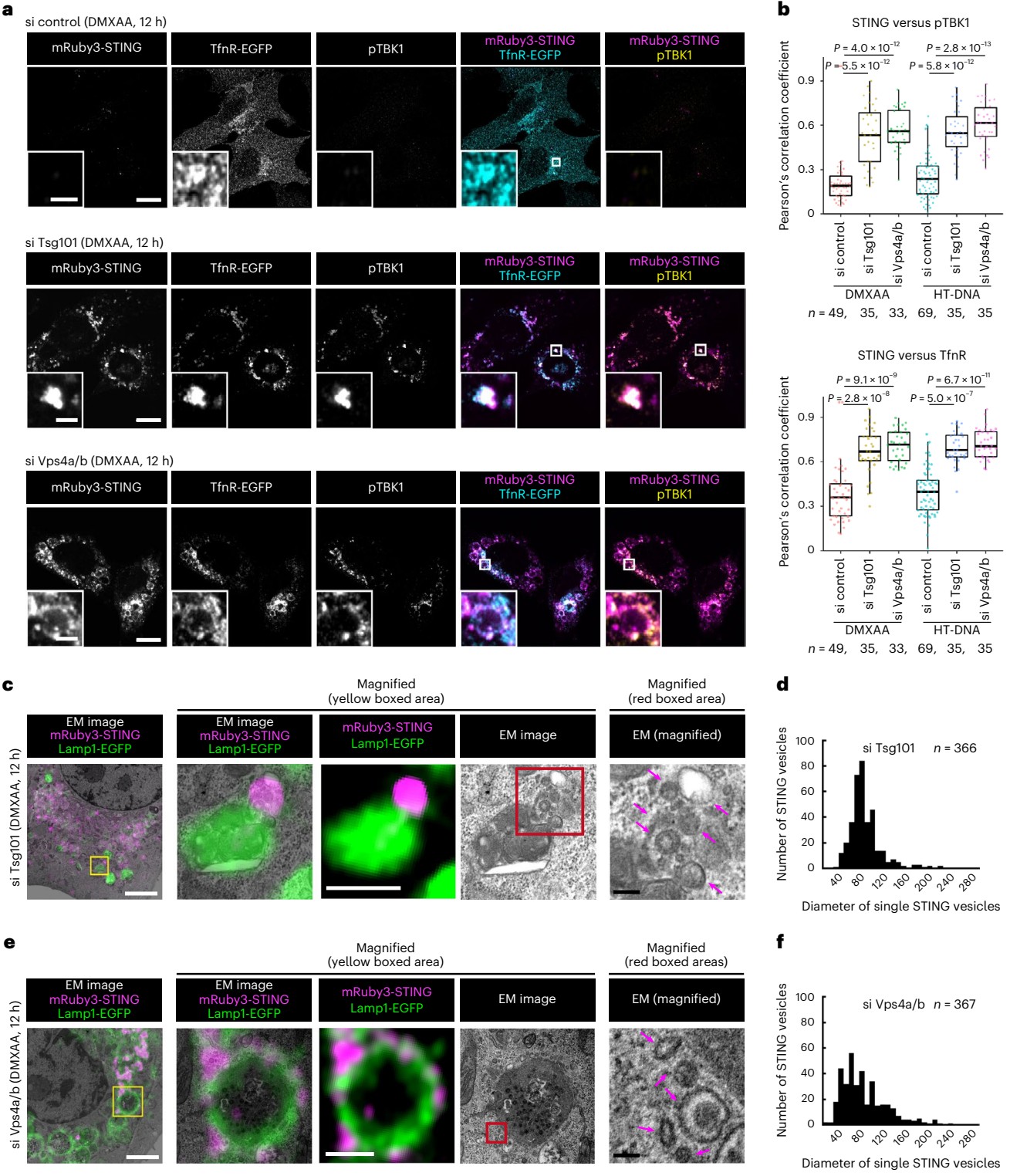

**Fig. 5 | ESCRT proteins are required for encapsulation of STING into the lumen of Lamp1-positive compartments. a**, TfnR-EGFP (cyan) and mRuby3-STING (magenta) were stably expressed in *Sting*[−/−] MEFs. Cells were treated with the indicated siRNAs, and then stimulated with DMXAA. Cells were immunostained with anti-pTBK1 (yellow) antibody. **b**, The Pearson's correlation coefficient between mRuby3-STING and pTBK1, or between mRuby3-STING and TfnR-EGFP in **a** is shown. Data are presented in box-and-whisker plots with the minimum, maximum, sample median and first versus third quartiles. **c**–**f**, CLEM analysis of STING-positive vesicles. *Sting*[−/−] MEFs stably expressing mRuby3-STING (magenta) and Lamp1-EGFP (green) were treated with siRNA against *Tsg101* (**c**,**d**) or *Vps4a/b* (**e**,**f**), and then stimulated with DMXAA. Lamp1-positive lysosomes and STING-positive membranes were identified by Airyscan super-resolution microscopy before processing for transmission EM to examine their ultrastructure (**c**,**e**). The yellow boxed areas in **c** and **e** are magnified in the right panels, respectively. The red boxed areas in EM images are magnified in the bottom right panels, respectively. STING-positive vesicles in **c** and **e** are indicated by magenta arrows. The diameters of STING-positive vesicles in Tsg101- or Vps4a/b-depleted cells were measured and plotted as histogram (**d** and **f**). Scale bars, 10 μm (**a**), 500 nm (magnified images in **a**), 1 μm (left CLEM images in **c** and **e**), 500 nm (fluorescence images in **c** and **e**), 100 nm (magnified EM images in **c** and **e**). The sample size (*n*) represents the number of cells (**b**) or vesicles (**d**,**f**). Source numerical data are available in source data.

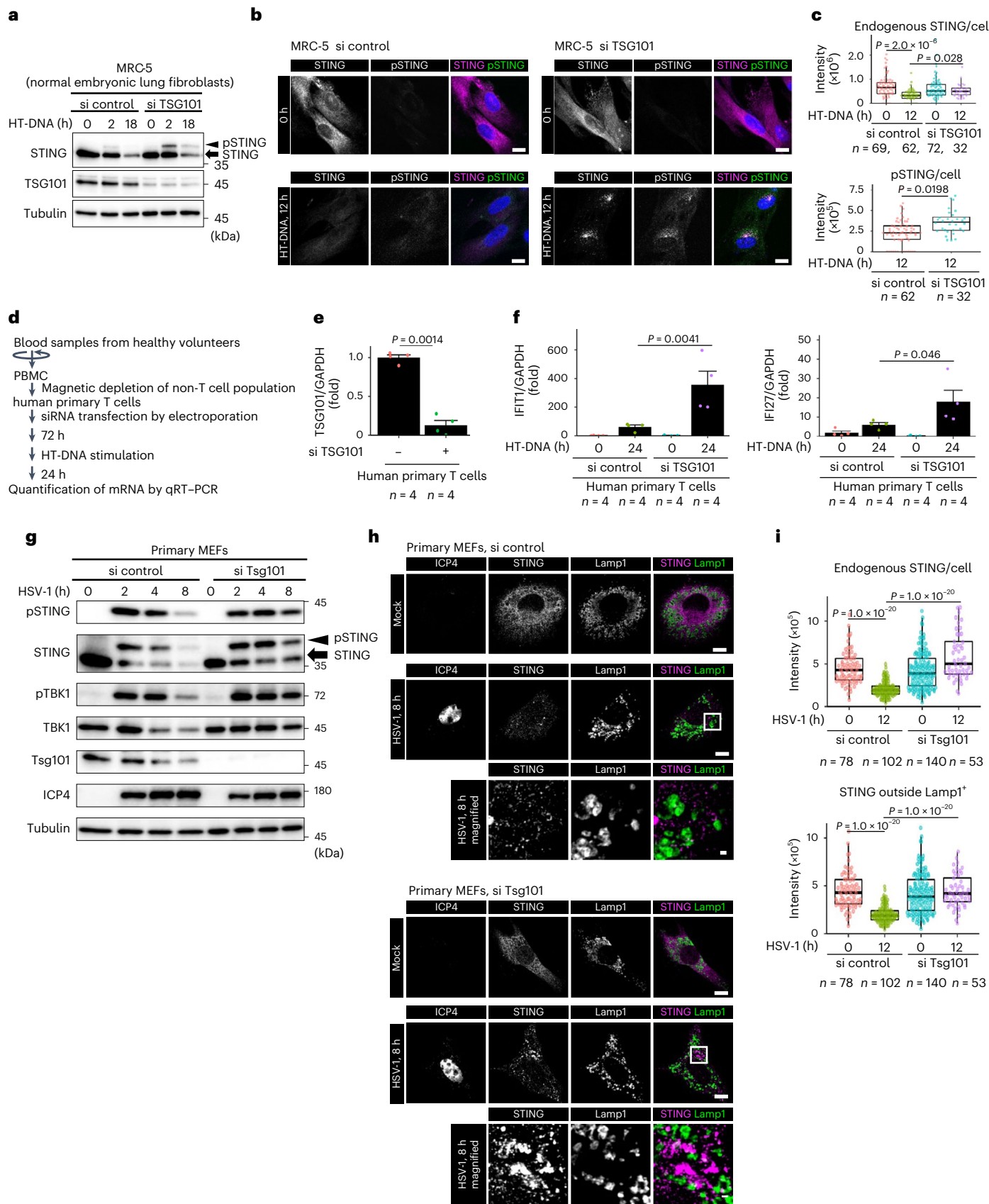

puncta 4 h after stimulation (Fig. 8f,g). In contrast, a Tsg101 variant lacking the UEV domain remained diffusive and did not translocate to the mRuby3-STING-positive puncta, suggesting that Tsg101 bound to ubiquitinated STING through its UEV domain. We then performed the split NanoLuc luciferase assay to examine the interaction between STING and Tsg101. As shown (Fig. 8h), we observed the stimulation-dependent interaction between STING and Tsg101 (WT), but not between STING and Tsg101 (ΔUEV).

**Fig. 6 | The physiological roles of Tsg101 in STING degradation and termination of type I interferon response. a**, MRC-5 cells were treated with siRNAs, and then stimulated with HT-DNA for the indicated times. Cell lysates were analysed by western blot. **b**, MRC-5 cells were treated with the indicated siRNAs, and then stimulated with HT-DNA. Cells were immunostained with anti-STING (magenta) and anti-pSTING (green) antibodies. **c**, The fluorescence intensity of STING or pSTING in **b** was quantified. **d**, Schematic representation of the experiments with human primary T cells. **e**, Knockdown efficiency of TSG101 gene in human primary T cells from a representative donor. Data are presented as mean ± s.d. **f**, The expression of IFIT1 or IFI27 was quantitated with qRT–PCR. Data are presented as mean ± s.d. **g**, Primary MEFs were treated with siRNAs, and then

infected with HSV-1 (MOI 10) for the indicated times. Cell lysates were analysed by western blot. **h**, Primary MEFs were treated with the indicated siRNAs, and then stimulated with HSV-1 infection (MOI 10) for 8 h. Cells were immunostained with anti-STING (magenta), anti-Lamp1 (green) and anti-ICP4 antibodies. **i**, The fluorescence intensity of STING in **h** was quantified. Scale bars, 10 μm (**b**,**h**) and 500 nm (magnified images in **h**). Data are presented in box-and-whisker plots with the minimum, maximum, sample median and first versus third quartiles (**c**,**i**). The sample size (n) represents the number of cells (**c**,**i**) or the biological replicates (**e**,**f**). Source numerical data and unprocessed blots are available in source data.

Knockdown of Tsg101 enhanced DMXAA-dependent induction of Cxcl10 (Figs. 4e and 8i). This enhanced transcription of Cxcl10 was suppressed by the expression of siRNA-resistant form of WT Tsg101, but not Tsg101 (ΔUEV) (Fig. 8i). mRuby3-STING lingered in Tsg101-depleted cells 12 h after stimulation (Fig. 8b,c). This duration of the fluorescence was suppressed by the expression of siRNA-resistant form of WT Tsg101, but not Tsg101 (ΔUEV) (Extended Data Fig. 10a,b). These results indicated that the binding of the UEV domain of Tsg101 to ubiquitinated STING was essential for lysosomal degradation of STING and the termination of the STING signalling.

## Discussion

The degradation of STING at lysosomes is pivotal to prevent the persistent transcription of innate immune and inflammatory genes[11,14]. However, the molecular machinery underlying the degradation of STING has not been clear. The stimulation of STING triggers macroautophagy[11,16], but this appears not to be involved in the degradation of STING[16] (Extended Data Fig. 1f–h). In the present study, we showed that STING vesicles originating from REs were directly encapsulated into the lumen of Lamp1+ in an ESCRT-dependent fashion and degraded (Fig. 8j and Extended Data Fig. 10f).

There is growing evidence that mammalian lysosomal microautophagy plays a role in the degradation of a variety of substrates, such as ER domains containing misfolded collagen[34] or the translocon complex[35], lipid droplet[36] or the transmembrane protein on the limiting membrane of lysosomes[37]. However, how these degradative substrates are recognized by lysosomes is not determined. In the present study, we showed that K63-linked polyubiquitin served as a degradation signal for lysosomal microautophagy. We further provided evidence that polyubiquitin of STING interacted with Tsg101 through its UEV domain, which may lead to the recruitment of other ESCRT proteins to complete lysosomal microautophagy. Of note, the very recent nuclear magnetic resonance result supported the direct binding of Tsg101 with K63-linked ubiquitin[38].

CMA is another way for lysosomes to digest cytosolic substances[21,39]. Mechanistically, KFERQ-like motifs present in the substrate proteins are recognized by a cytosolic chaperone protein Hsc70c and directed to Lamp2A at lysosomal surface, followed by the translocation of the substrate proteins through lysosomal membrane. CMA is not expected to mediate the degradation of transmembrane proteins. In line with this, we confirmed that knockdown of Lamp2 did not interfere with the encapsulation of STING into lysosomes or the expression of stimulation-dependent transcription of Cxcl10 (Extended Data Fig. 10c,d). We also found that STING stimulation did not cause the noticeable degradation of Gapdh, an authentic substrate of CMA (Extended Data Fig. 10e). Thus, the contribution of CMA to STING degradation was ruled out.

By screening of mammalian *Vps* genes, we identified several ESCRT proteins as essential regulators of lysosomal microautophagy. The ESCRT generates inverse membrane involutions on a variety of organellar membranes and cooperates with the ATPase Vps4 to drive membrane scission or sealing. On early endosomes/late endosomes, the ESCRT plays a key role in the biogenesis of intraluminal vesicles (ILVs), which are destined for degradation at lysosomes or for extracellular secretion as exosomes. Intriguingly, the size of the endosomal ILVs is rather small, ranging between 40 and 80 nm (ref. [40]), which highly contrasts with that of lysosomal ILVs (indicated by orange arrowheads in Fig. 3b,c), ranging between 200 and 300 nm. Therefore, the nature of the ESCRT operating on lysosomes may be distinct from that on endosomes. In this regard, it is noteworthy that a component of ESCRT-0, Hgs (also known as Hrs), was dispensable for STING degradation (Extended Data Fig. 7a).

Infection of *Listeria monocytogenes* activates the GAS/STING pathway. Intriguingly, activated TBK1 phosphorylates MVB12b, a subunit of ESCRT-I. The phosphorylation of MVB12b is essential for sorting *Listeria* DNA into ILVs, which are destined for extracellular secretion as exosomes[41]. We expect that ubiquitination of STING not only functions in the recruitment of Tsg101, but may affect the assembly and/or function of the ESCRT, so that the ESCRT can function on lysosomes.

**Fig. 7 | Ubiquitination on K288 of STING is required for STING degradation and termination of type I interferon response. a**, *Sting*−/− MEFs reconstituted with EGFP-STING were stimulated with DMXAA for the indicated times. EGFP-STING was immunoprecipitated with anti-GFP antibody. The cell lysates and the immunoprecipitated proteins were analysed by western blot. IP, immunoprecipitation. **b**, *Sting*−/− MEFs stably expressing mRuby3-STING and mNeonGreen (mNG)-ubiquitin were imaged every 5 min after DMXAA stimulation. **c**, Quantitation of the number of mNG-ubiquitin puncta (see also Supplementary Video 3). **d**, *Sting*−/− MEFs reconstituted with EGFP-STING (WT, K19R, K150/151R, K235R, K288R or K337R) were stimulated with DMXAA. EGFP-STING was immunoprecipitated with anti-GFP antibody. The cell lysates and the immunoprecipitated proteins were analysed by western blot. **e**, The fluorescence intensity of EGFP-STING (WT or K288R) under the indicated conditions was quantified. NS, not significant. **f**, *Sting*−/− MEFs reconstituted with EGFP-STING (WT or K288R) were stimulated with DMXAA. Cells were immunostained with anti-GM130 or anti-Rab11 antibodies. The Pearson's correlation coefficient between EGFP-STING (WT or K288R) and GM130, or between EGFP-STING

(WT or K288R) and Rab11, is shown. **g**, Cells were stimulated with DMXAA. Cell lysates were analysed by western blot. The band intensities were quantified. [STING/tubulin], [pTBK1/TBK1] and [pIRF3/IRF3] were calculated. **h**, Cells were stimulated with DMXAA or HT-DNA for 12 h. The expression of Cxcl10 was quantitated with qRT–PCR. Data are presented as mean ± s.d. **i**, *Sting*−/− MEFs reconstituted with EGFP-STING (WT or K288R) were stimulated with DMXAA. Cells were immunostained with anti-K63 ubiquitin antibody. **j**, The Pearson's correlation coefficient between EGFP-STING (WT or K288R) and K63 ubiquitin is shown. **k**, Cells were stimulated with DMXAA. Cell lysates were prepared, and EGFP-STING was immunoprecipitated with anti-GFP antibody. The cell lysates and the immunoprecipitated proteins were analysed by western blot. Scale bars, 10 μm (**b**,**f**,**i**) and 500 nm (magnified images in **b** and **i**). Data are presented in box-and-whisker plots with the minimum, maximum, sample median and first versus third quartiles (**e**,**f**,**j**). The sample size (n) represents the number of cells (**e**,**f**,**j**) or the biological replicates (**h**). Source numerical data and unprocessed blots are available in source data.

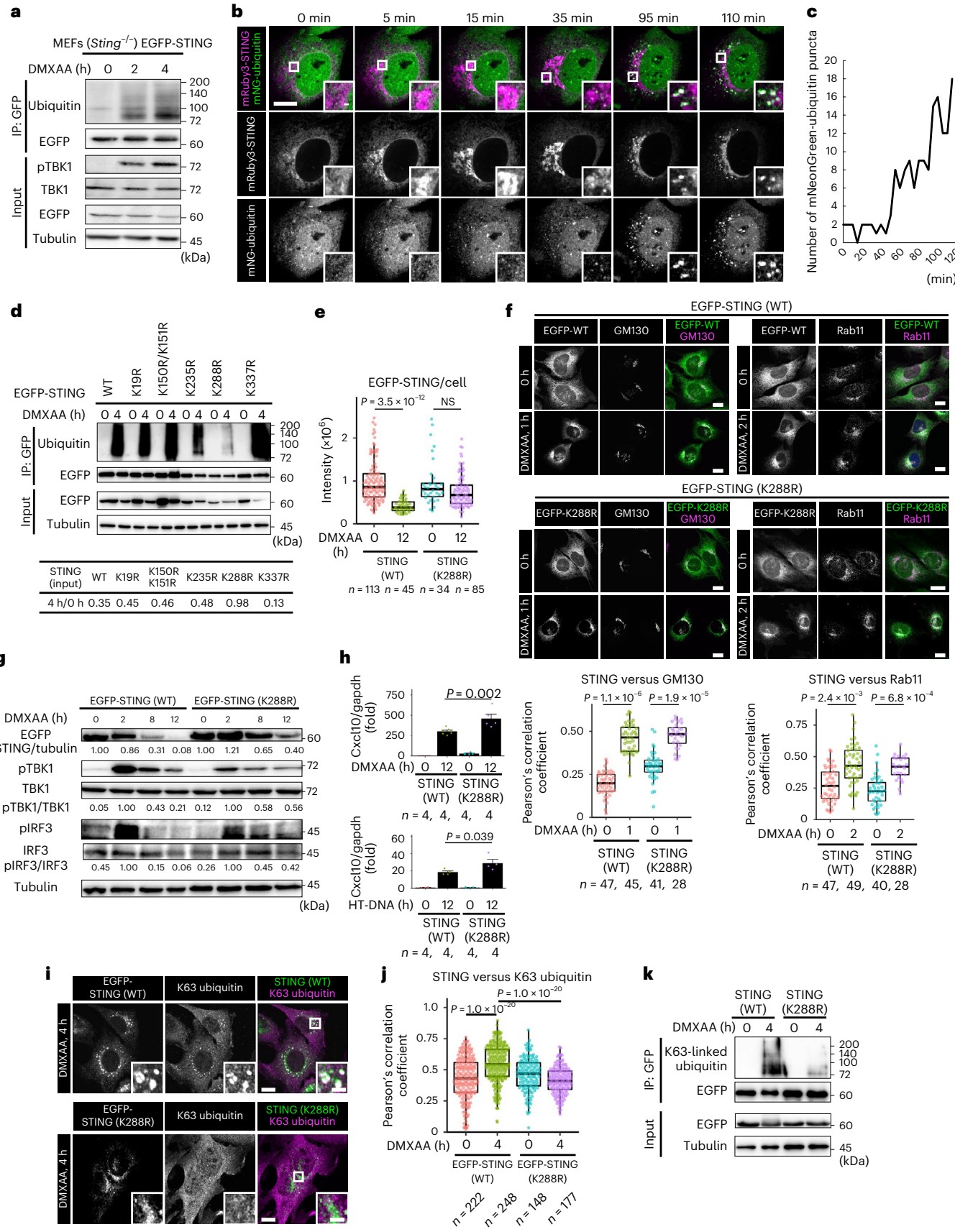

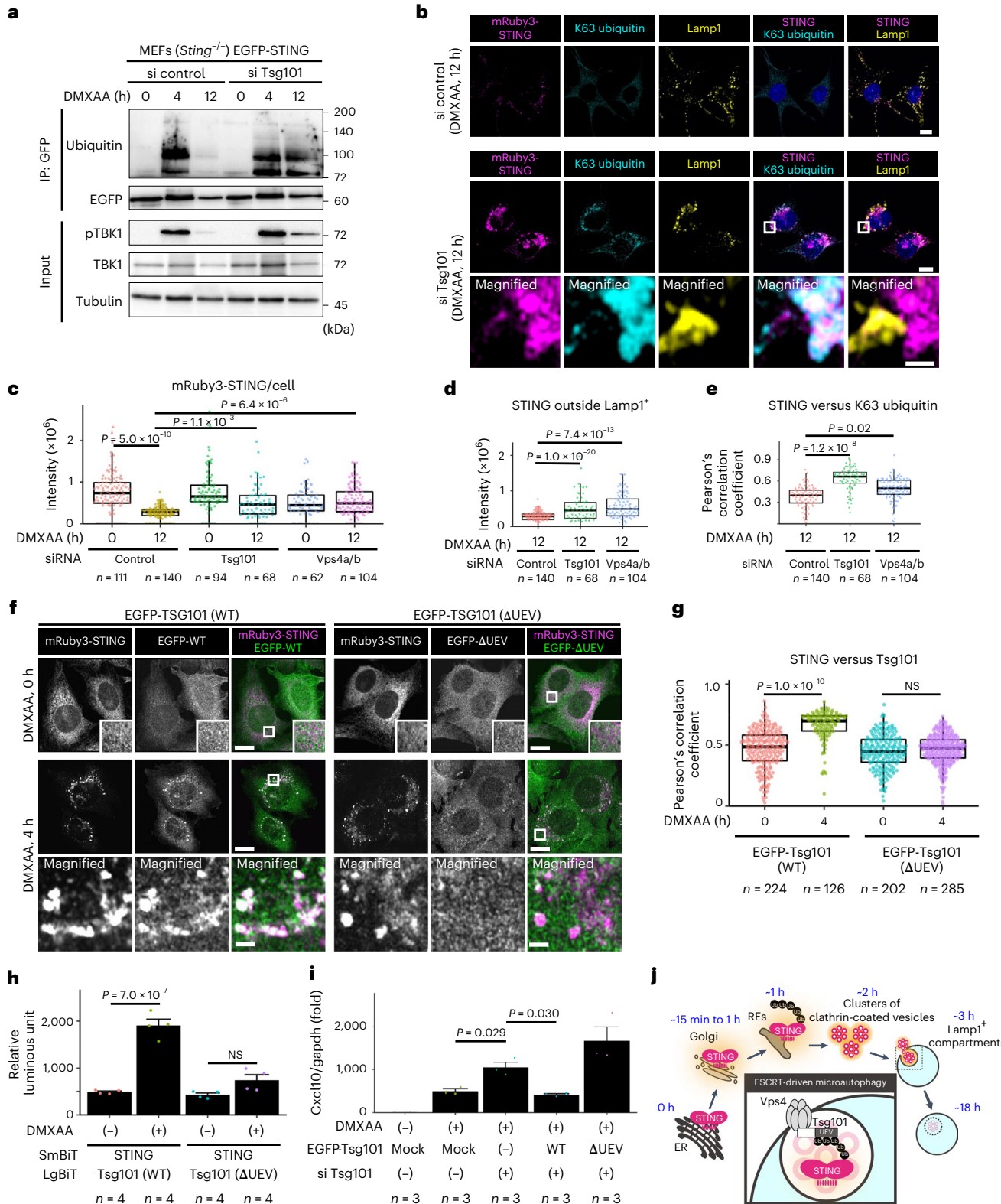

During or after STING stay at REs, STING was located on a cluster of uniform membrane vesicles with electron-dense coat (Fig. 3d,e), which appeared as clathrin-coated vesicles (Fig. 3j). The vesiculation of STING membrane, through the reduction of its size, may facilitate the process of lysosomal encapsulation. Coat proteins may endow STING membranes with stiffness so that lysosomal encapsulation would proceed efficiently. Given that STING underwent extensive ubiquitination

at the Golgi/REs (Fig. 7 and Extended Data Fig. 9), this ubiquitination may be coupled to the packaging of STING into clathrin-coated vesicles.

REs are organelles that function in recycling molecules back to the plasma membrane[17,42]. Besides their classical roles in endocytic recycling, it has been shown that REs have a role in exocytic and retrograde membrane traffic[43,44], demonstrating that REs serve as a central hub for sorting various cargos to different destinations[45]. Of note, clathrin

**Fig. 8 | Ubiquitin-binding domain of Tsg101 is required for STING degradation and termination of type I interferon response. a**, *Sting*⁻/⁻ MEFs reconstituted with EGFP-STING were treated with control siRNA or Tsg101 siRNA. Cells were then incubated with DMXAA. EGFP-STING was immunoprecipitated with anti-GFP antibody. The cell lysates and the immunoprecipitated proteins were analysed by western blot. **b**, *Sting*⁻/⁻ MEFs reconstituted with mRuby3-STING were treated with control siRNA or *Tsg101* siRNA, and then stimulated with DMXAA. Cells were immunostained with anti-K63 ubiquitin antibody (cyan) and anti-Lamp1 (yellow). The boxed areas are magnified in the bottom row. **c**, *Sting*⁻/⁻ MEFs reconstituted with mRuby3-STING were treated with indicated siRNAs, and then stimulated with DMXAA. The fluorescence intensity of mRuby3-STING under the indicated conditions was quantified. **d**, The fluorescence intensity of mRuby3-STING that was not associated with Lamp1⁺ in **b** was quantified. **e**, The Pearson's correlation coefficient between mRuby3-STING and K63 ubiquitin in **b**

is shown. **f**, EGFP-Tsg101 (WT or ΔUEV) and mRuby3-STING were stably expressed in *Sting*⁻/⁻ MEFs. Cells were treated with DMXAA. The boxed areas are magnified in the bottom row. **g**, The Pearson's correlation coefficient between mRuby3-STING and EGFP-Tsg101 (WT or ΔUEV) in **f** is shown. NS, not significant. **h**, LgBiT-Tsg101 (WT or ΔUEV) and SmBiT-STING were stably expressed in *Sting*⁻/⁻ MEFs. Cells were treated with DMXAA for 4 h. Data are presented as mean ± s.d. NS, not significant. **i**, The expression of Cxcl10 was quantitated with qRT–PCR. Data are presented as mean ± s.d. **j**, A graphical abstract illustrating ESCRT-driven microautophagy. Scale bars, 10 μm (**b**,**f**) and 500 nm (magnified images in **b** and **f**). Data are presented in box-and-whisker plots with the minimum, maximum, sample median and first versus third quartiles (**c**–**e**,**g**). The sample size (*n*) represents the number of cells for (**c**–**e**,**g**) or the biological replicates (**h**,**i**). Source numerical data and unprocessed blots are available in source data.

and clathrin adaptor AP-1 function at REs for the retrograde membrane traffic[46]. In the present study, we revealed that REs also had a role in a previously unanticipated traffic pathway by which an exocytic cargo protein STING was delivered to lysosomes. Given the nature of the pathway that STING follows[47,48], namely, 'ER–Golgi–REs–lysosomes', lysosomal microautophagy may contribute to the proteostasis of exocytic proteins and ER/Golgi resident proteins.

## Online content

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

Yoshihiko Kuchitsu[1,11], Kojiro Mukai ®[1,11], Rei Uematsu[2], Yuki Takaada[1], Ayumi Shinojima[1], Ruri Shindo[1], Tsumugi Shoji[1], Shiori Hamano[1], Emari Ogawa[2], Ryota Sato[3], Kensuke Miyake ®[3], Akihisa Kato[4,5], Yasushi Kawaguchi ®[4,5], Masahiko Nishitani-Isa[6], Kazushi Izawa ®[6], Ryuta Nishikomori[7], Takahiro Yasumi ®[6], Takehiro Suzuki[8], Naoshi Dohmae ®[8], Takefumi Uemura[9], Glen N. Barber[10], Hiroyuki Arai ®[2], Satoshi Waguri ®[9]✉ & Tomohiko Taguchi ®[1]✉

[1]Laboratory of Organelle Pathophysiology, Department of Integrative Life Sciences, Graduate School of Life Sciences, Tohoku University, Sendai, Japan. [2]Department of Health Chemistry, Graduate School of Pharmaceutical Sciences, University of Tokyo, Tokyo, Japan. [3]Division of Innate Immunity, Department of Microbiology and Immunology, The Institute of Medical Science, The University of Tokyo, Tokyo, Japan. [4]Division of Molecular Virology, Department of Microbiology and Immunology, The Institute of Medical Science, The University of Tokyo, Tokyo, Japan. [5]Department of Infectious Disease Control, International Research Center for Infectious Diseases, The Institute of Medical Science, The University of Tokyo, Tokyo, Japan. [6]Department of Pediatrics, Kyoto University Graduate School of Medicine, Kyoto, Japan. [7]Department of Pediatrics and Child Health, Kurume University School of Medicine, Kurume, Japan. [8]Biomolecular Characterization Unit, RIKEN Center for Sustainable Resource Science, Wako, Japan. [9]Department of Anatomy and Histology, Fukushima Medical University School of Medicine, Fukushima, Japan. [10]Department of Cell Biology and Sylvester Comprehensive Cancer Center, University of Miami School of Medicine, Miami, FL, USA. [11]These authors contributed equally: Yoshihiko Kuchitsu, Kojiro Mukai. ✉e-mail: waguri@fmu.ac.jp; tom_taguchi@tohoku.ac.jp

## Methods

### Ethical approval

MEFs from mice were collected according to ethics number PA17-84 approved by the Institute of Medical Sciences of the University of Tokyo. All experiments involving human subjects were conducted in accordance with the principles of the Declaration of Helsinki and were approved by the ethics committee of Kyoto University Hospital (protocol number G1233). Written informed consent was obtained from the participants before sampling. No compensation was provided.

### Reagents

Antibodies used in this study are shown in Supplementary Table 2. The following reagents were purchased from the manufacturers as noted: DMXAA (14617, Cayman), anti-FLAG M2 Affinity Gel (A2220, Sigma), LysoTracker Deep Red (L12492, Thermo Fisher Scientific), E64d (4321, Peptide Institute), pepstatin A (4397, Peptide Institute), orlistat (O4139, MERCK) and HT-DNA (D6898, Sigma).

### Cell culture

MEFs were obtained from embryos of WT or *Sting*$^{-/-}$ mice at E13.5 and immortalized with SV40 Large T antigen. MEFs were cultured in DMEM supplemented with 10% foetal bovine serum (FBS) and penicillin/streptomycin/glutamine (PSG) in a 5% $CO_2$ incubator. MEFs that stably express tagged proteins were established using retrovirus. Plat-E cells were transfected with pMXs vectors, and the medium that contains the retrovirus was collected. MEFs were incubated with the medium and then selected with puromycin (2 µg ml$^{-1}$), blasticidin (5 µg ml$^{-1}$) or hygromycin (400 µg ml$^{-1}$) for several days. RAW-Lucia ISG-KO-STING Cells (InvivoGen) were cultured in DMEM supplemented with 10% FBS, normocin (100 µg ml$^{-1}$) and PSG. MRC-5 cells, human normal embryonic lung fibroblasts, were obtained from the Riken BioResource Center. Simian kidney epithelial Vero cells were provided by Dr Bernard Roizman and maintained in DMEM containing 5% calf serum.

### Immunocytochemistry

Cells were seeded on coverslips (13 mm No.1 S, MATSUNAMI), fixed with 4% paraformaldehyde (PFA) in PBS at room temperature for 15 min, and permeabilized with digitonin (50 µg ml$^{-1}$) in PBS at room temperature for 5 min. After blocking with 3% BSA in PBS, cells were incubated with primary antibodies followed by secondary antibodies at room temperature for 1 h. When necessary, cells were stained with DAPI and HCS CellMask Deep Red Stain (H32721, Thermo Fisher Scientific) for segmentation of cells. Cells were then mounted with ProLong Glass Antifade Mountant (P36982, Thermo Fisher Scientific).

Confocal microscopy was performed using LSM880 with Airyscan (Zeiss) with 20 × 0.8 Plan-Apochromat dry lens, 63 × 1.4 Plan-Apochromat oil immersion lens or 100 × 1.46 alpha-Plan-Apochromat oil immersion lens. Images were analysed and processed with Zeiss ZEN 2.3 SP1 FP3 (black, 64 bit) (version 14.0.21.201) and Fiji (version 2.1.0/1.53c).

### Live-cell imaging

The day before imaging, cells were seeded on a glass-bottom dish (627870, Greiner Bio-One). The medium was changed to DMEM$^{gfp}$-2 (MC102, Evrogen) containing 10% FBS, PSG and rutin (20 µg ml$^{-1}$) (30319-04, Nacalai Tesque) before imaging. HaloTag SaraFluor 650T ligand was added to the medium for 10 min before live-cell imaging to visualize HaloTag-conjugated protein. Live-cell imaging was performed using LSM880 with Airyscan (Zeiss) equipped with a 100 × 1.46 alpha-Plan-Apochromat oil immersion lens and Immersol 518 F/37 °C (444970-9010-000, Zeiss). During live-cell imaging, the dish was mounted in a chamber (STXG-WSKMX-SET, TOKAI HIT) to maintain the incubation conditions at 37 °C and 5% $CO_2$. Acuired images were Airyscan processed with Zeiss ZEN 2.3 SP1 FP3 (black, 64 bit) (version 14.0.21.201) and analysed with Fiji (version 2.1.0/1.53c).

### PCR cloning

Complementary DNAs (CDNAs) encoding mouse STING, mouse Rab5a, mouse TfnR, human Lamp1, mouse ubiquitin and mouse Tsg101 were amplified by PCR. The cDNAs were inserted into pMX-IPuro or pMX-IBla. Tsg101 (ΔUEV (aa 146-391)) and siRNA-resistant Tsg101 were generated by site-directed mutagenesis.

### Type I interferon bioassay

MEFs were treated with indicated siRNA for 62 h followed by stimulation with DMXAA for 10 h. Cell culture supernatants were then added to Raw264.7-Lucia ISG-KO-STING Cells (Invivogen). Twelve hours after incubation, the luciferase activity was measured by GloMax Navigator Microplate Luminometer (Promega) (version 3.1.0).

### Flow cytometry

*Sting*$^{-/-}$ MEFs reconstituted with mRuby3-STING were treated with indicated siRNA for 54 h followed by stimulation with or without DMXAA for 18 h. Cells were detached with trypsin/EDTA and fixed with 4% PFA in PBS at room temperature for 15 min. Mean fluorescence intensity (MFI) was analysed by Cell Sorter SH800 (Sony).

### qRT–PCR

Total RNA was extracted from cells using ISOGEN II (Nippongene) or SuperPrep II (TOYOBO), and reverse-transcribed using ReverTraAce qPCR RT Master Mix with gDNA Remover (TOYOBO). Quantitative real-time PCR (qRT–PCR) was performed using KOD SYBR qPCR (TOYOBO) and LightCycler 96 (Roche). Target gene expression was normalized on the basis of Gapdh content.

### Immunoprecipitation

Cells were washed with ice-cold PBS and scraped in immunoprecipitation buffer composed of 50 mM HEPES−NaOH (pH 7.2), 150 mM NaCl, 5 mM EDTA, 1% Triton X-100, protease inhibitor cocktail (25955, dilution 1:100) (Nacalai Tesque) and phosphatase inhibitors (8 mM NaF, 12 mM β-glycerophosphate, 1 mM $Na_3VO_4$, 1.2 mM $Na_2MoO_4$, 5 mM cantharidin and 2 mM imidazole). The cell lysates were centrifuged at 20,000$g$ for 15 min at 4 °C, and the resultant supernatants were pre-cleared with Ig-Accept Protein G (Nacalai Tesque) at 4 °C for 15 min. The lysates were then incubated for 3 h at 4 °C with anti-GFP (3E6) and Ig-Accept Protein G. The beads were washed four times with immunoprecipitation wash buffer (50 mM HEPES−NaOH (pH 7.2), 150 mM NaCl and 0.1% Triton X-100) and eluted with 2× Laemmli sample buffer. The immunoprecipitated proteins were separated with SDS−PAGE and transferred to polyvinylidene difluoride membrane, then analysed by western blot.

### Western blotting

Proteins were separated in polyacrylamide gel and then transferred to polyvinylidene difluoride membranes (Millipore). These membranes were incubated with primary antibodies, followed by secondary antibodies conjugated to peroxidase. The proteins were visualized by enhanced chemiluminescence using Fusion SOLO.7S. EDGE (Vilber-Lourmat).

### MS

Cells were lysed with immunoprecipitation buffer (50 mM HEPES−NaOH (pH 7.2), 150 mM NaCl, 5 mM EDTA, 1% Triton X-100, protease inhibitors and phosphatase inhibitors). The lysates were centrifuged at 20,000$g$ for 10 min at 4 °C, and the resultant supernatants were incubated for overnight at 4 °C with anti-FLAG M2 Affinity Gel. The beads were washed four times with immunoprecipitation wash buffer (50 mM HEPES−NaOH (pH 7.2), 150 mM NaCl and 1% Triton X-100), and eluted with elution buffer (50 mM HEPES−NaOH (pH 7.2), 150 mM NaCl, 5 mM EDTA, 1% Triton X-100 and 500 µg ml$^{-1}$ FLAG peptide). Eluted proteins were applied to SDS−PAGE, and the electrophoresis was stopped when the samples were moved to the top of the separation gel. The gel was

stained with Coomassie brilliant blue and the protein bands at the top of separation gel were excised. The proteins were reduced and alkylated with acrylamide, followed by a tryptic digestion in gel (TPCK treated trypsin, Worthington Biochemical Corporation). The digests were separated with a reversed phase nano-spray column (NTCC-360/75-3-105, Nikkyo Technos) and then applied to Q Exactiv Hybrid Quadrupole-Orbitrap mass spectrometer (Thermo Scientific). Mass spectrometry (MS) and tandem MS (MS/MS) data were obtained with TOP10 method. The MS/MS data were searched against the National Center for Biotechnology Information nr database (https://www.ncbi.nlm.nih.gov) using MASCOT program 2.6 (Matrix Science), and the MS data were quantified using Proteome Discoverer 2.2 (Thermo Scientific). MS data have been deposited in ProteomeXchange with the primary accession code PXD039411.

### RNA interference

siRNA (siGENOME) used in this study was purchased from Dharmacon. Cells were transfected with siRNA (5 nM) using Lipofectamine RNAiMAX (Invitrogen) according to the manufacturer's instruction. Six hours after transfection, the medium was replaced by DMEM with 10% FBS followed by incubation for 66 h.

### Quantification of imaging data

For quantification of imaging data of multiple cells, individual cells were segmented by Cellpose[49], a deep learning-based segmentation method with cytosol and nucleus images. Pearson's correlation coefficient was quantified by BIOP JACoP in Fiji plugin with region of interest (ROI) data from Cellpose.

The signal intensity of STING in each whole cell was quantified by using ROI of the cell. Lysosome areas were extracted and binarized from Lamp1 image by Trainable Weka Segmentation, a machine learning tool for microscopy pixel classification in Fiji plugin. STING image within lysosomes was then extracted by multiplying the binarized Lamp1 image by STING image. The signal intensity of STING inside lysosomes in each cell was quantified with ROI of the cell and the extracted STING image. The signal intensity of STING outside lysosomes was quantified by subtracting the intensity inside lysosomes from that of the whole cell.

To quantify the number of mNeonGreen-ubiquitin puncta, images of mNeonGreen-ubiquitin were thresholded using Yen's method with Fiji. mNeonGreen-ubiquitin positive puncta were defined using the 'Analyze Particles' menu from Fiji on the binary thresholded image.

### CLEM

Cells were cultured on coverslips coated with 150 μm grids (Matsunami Glass Ind.). The cells were stimulated with DMXAA (25 μg ml⁻¹) in the presence of protease inhibitors (E64d (30 μg ml⁻¹) and pepstatin A (40 μg ml⁻¹)) and orlistat (20 μg ml⁻¹). Cells were fixed with 2% PFA–2% glutaraldehyde in 0.1 M phosphate buffer (pH 7.4) for 15 min at room temperature and rinsed three times for 15 min each time in 0.1 M phosphate buffer (pH 7.4). The fluorescence images were obtained using a confocal microscope (LSM880 with Airyscan (Zeiss)). They were fixed again with 2% PFA–2% glutaraldehyde in 0.1 M phosphate buffer (pH 7.4) for more than 15 min at 4 °C, and then with a reduced osmium fixative. After embedding in Epon812 resin, areas containing cells of interest were trimmed according to the light-microscopic observations, and serial ultrathin sections (80 nm thickness) were prepared and observed with an electron microscope (JEM1400EX; JEOL)[50,51].

### Split NanoLuc luciferase assay

*Sting*⁻/⁻ MEFs stably expressing LgBiT-STING and SmBiT-Tsg101 (WT or ΔUEV) were treated with Nano-Glo Endurazine substrate (N2570, Promega) for 2 h at 37 °C. Cells were then stimulated with vehicle (DMSO) or DMXAA for 4 h. The luciferase activity was measured by GloMax Navigator Microplate Luminometer (Promega) (version 3.1.0).

### Preparation of virus and virus infection

Recombinant HSV-1 R3616 (ref. [52]) in which a 1 kb fragment from the coding region of the γ34.5 gene was deleted was kindly provided by Bernard Roizman. Vero cells infected with the recombinant HSV-1 at a multiplicity of infection (MOI) of 0.01 for 48 h were collected by low-speed centrifugation. After freeze–thawing, lysates were briefly sonicated on ice and clarified by low-speed centrifugation, and the supernatant was passed through 0.45-μm-pore-size filters. The virus-containing supernatant was layered onto a 28 ml discontinuous sucrose gradient (60% and 30%) in PBS and centrifuged for 90 min at 146,000 *g* in a P32ST swing rotor (Eppendorf Himac Technologies) to produce a visible band of viruses. Purified viruses were then collected, pelleted by centrifugation for 90 min at 146,000 *g* in a P32ST swing rotor through a 30% sucrose cushion, and resuspended in a small volume of PBS. Purified viruses were stored at −80 °C.

Primary MEFs were seeded on Cellmatrix TYPE I-A coated coverslips and transfected with siRNA using Lipofectamine RNAiMAX (Invitrogen). Two days after transfection, cells were infected with the HSV-1 at an MOI of 10. Cells were then fixed or lysed for immunocytochemistry or western blot, respectively.

### mRNA silencing and HT-DNA stimulation of human T cells

Blood samples were collected from four Japanese males aged 30–55 years with no significant medical history. Peripheral blood mononuclear cells (PBMCs) were prepared by density gradient centrifugation of whole blood samples, and T cells were isolated magnetically using Pan T Cell Isolation Kit and autoMACS Pro Separator (Miltenyi Biotec) according to the manufacturer's instructions. T cells were transfected with 500 nM siRNAs using 4D-Nucleofector and the P3 Primary Cell 4D-Nucleofector X Kit (Lonza). Seventy-two hours after electroporation, cells were stimulated with HT-DNA (2 μg ml⁻¹) using the Lipofectamine 2000 transfection reagent (Thermo Fisher Scientific) and collected for analyses 24 h later.

### Statistics and reproducibility

Error bars displayed in bar plots throughout this study represent standard error of the mean unless otherwise indicated and were calculated from triplicate or quadruplicate samples. In box-and-whisker plots, the box bounds the interquartile range divided by the median, and whiskers extend to a maximum of 1.5× interquartile range beyond the box. The corresponding data points are overlaid on the plots. The data were statistically analysed by performing Student's unpaired two-tailed *t*-test with Bonferroni multiple correction (Figs. 4e and 8i), one-way analysis of variance followed by Tukey–Kramer post hoc test for multiple comparisons (Figs. 1b,c,g–i, 3i, 5b, 6c,e,i, 7e,j and 8c–e,g,h and Extended Data Figs. 1e,h, 4b, 3c,d, 6b and 9d–f), or Dunnett's test for multiple comparisons (Figs. 6f and 7f,h) with R (version 4.1.2) and KNIME (version 4.5.1). No statistical method was used to pre-determine sample size. No data were excluded from the analyses. The experiments were not randomized.

### Reporting summary

Further information on research design is available in the Nature Portfolio Reporting Summary linked to this article.

## Data availability

MS data have been deposited in ProteomeXchange with the primary accession code PXD039411. The datasets generated in the current study are included in the supplementary information. Source data are provided with this paper. All other data supporting the findings of this study are available from the corresponding author on reasonable request.

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

## Acknowledgements

This work was supported by JSPS KAKENHI grant numbers JP19H00974 (T.T.), JP17H06164 (H.A.), JP17H06418 (H.A.), JP20H03415 (S.W.), JP20H05307 (K.Mu), JP20H03202 (K.Mu), JP17K15445 (K.Mu), JP21K07104 (T.U.), JP21K15464 (R.Sa) and JP20H05692 (Y.Ka); AMED-PRIME (17939604) (T.T.), AMED (JP19ek0109387) (R.N.); JST CREST (JPMJCR21E4) (K.Mu, R.Sa and K.Mi), JST Center of Innovation programme from Japan (JPM-JCE1303) (K.Mu); JSPS Research Fellowship for Young Scientists (19J21426, Y.Ku; 19J23315, E.O.); the Subsidy for Interdisciplinary Study and Research concerning COVID-19 (Mitsubishi Foundation) (T.T.), Takeda Science Foundation (T.T., S.W. and K.Mu), the Cell Science Research Foundation (K.Mu), Grant for Basic Science Research Projects from the Sumitomo Foundation (K.Mu), Koyanagi-Foundation (K.Mu), the Nakatomi Foundation (K.Mu), SGH Cancer Research Grant (K.Mu) and Research Grant of the Princess Takamatsu Cancer Research Fund (K.Mu). We thank T. Yabe and K. Kanno for their technical support in the EM. We thank N. Mizushima for Atg5 tet-off MEFs, H. Konno for technical support in experimental design with HSV-1, and A. Inoue for technical support in split NanoLuc luciferase assay. We thank C. Kurata and M. Higashiguchi for technical support in experiments with human primary T cells.

## Author contributions

Y.Ku and K.Mu designed and performed the experiments, analysed the data, interpreted the results and wrote the paper; R.U. analysed STING ubiquitination and performed experiments for identification of STING binding proteins; Y.T. performed siRNA screening; A.S. performed live-cell imaging; R.Sh. and T.Sh. performed the experiments with NanoLuc luciferase; S.H. performed qPCR analysis; E.O. designed and performed experiments; R.Sa and A.K. performed the experiments with HSV-1; K.Mi and Y.Ka designed the experiments with HSV-1; M.N.-I. performed the experiments with human primary T cells; K.I., R.N. and T.Y. designed the experiments with human primary T cells; T.Su. and N.D. performed the proteomics analysis; T.U. performed the experiments with EM; G.N.B. discussed the results; H.A. designed the experiments and interpreted the results; S.W. performed the experiments with EM and interpreted the results; T.T. designed the experiments, interpreted the results and wrote the paper.

## Competing interests

The authors declare no competing interests.

## Additional information

**Extended data** is available for this paper at https://doi.org/10.1038/s41556-023-01098-9.

**Correspondence and requests for materials** should be addressed to Satoshi Waguri or Tomohiko Taguchi.

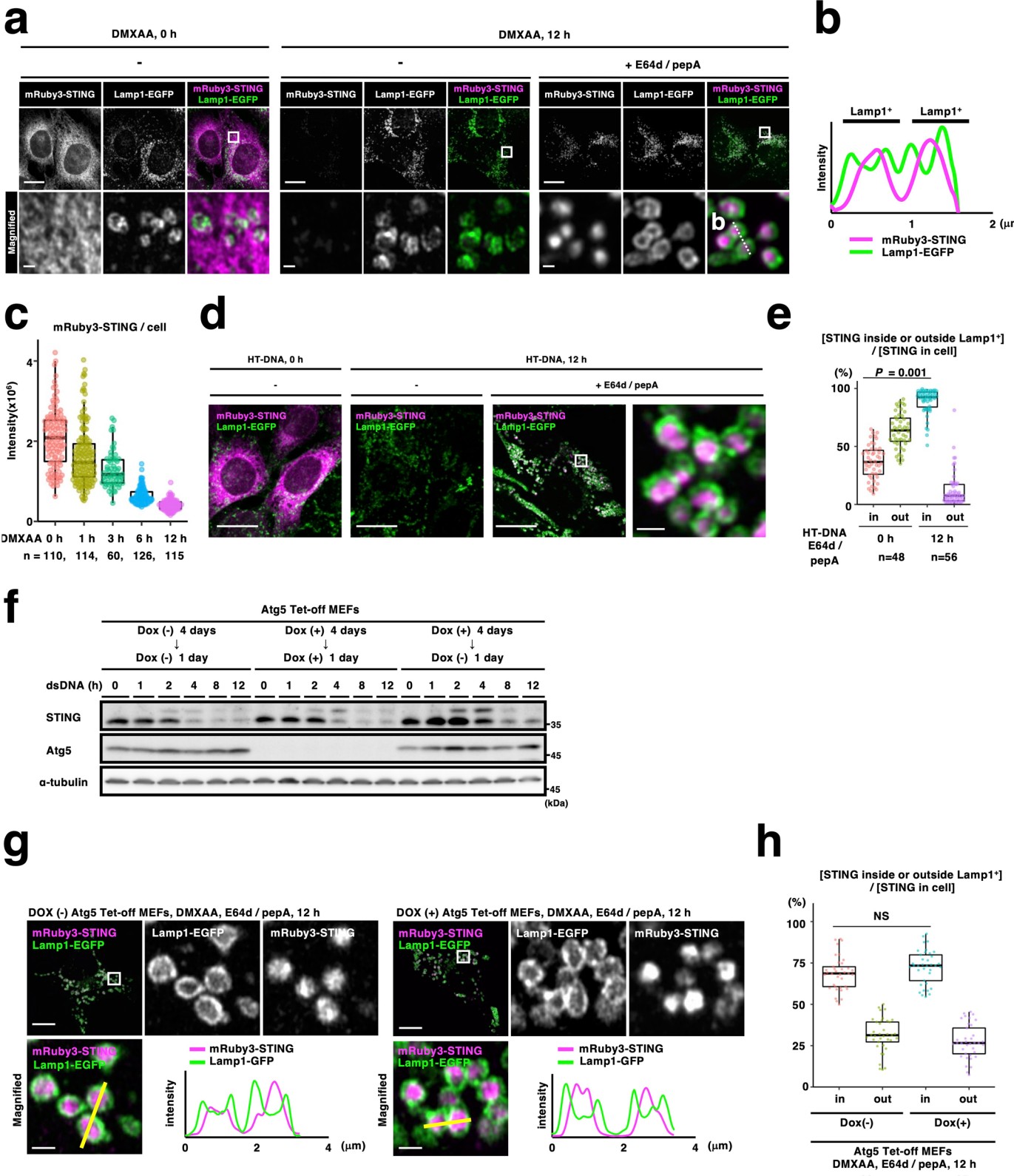

**Extended Data Fig. 1 | See next page for caption.**

**Extended Data Fig. 1 | Atg5-dependent macroautophagy was not involved in STING degradation. a**, *Sting*[−/−] MEFs stably expressing mRuby3-STING and Lamp1-EGFP were treated with DMXAA for 12 h. For the inhibition of lysosomal proteolysis, E64d and pepstatin A were added to the medium. The boxed areas in the top panels are magnified in the bottom panels. **b**, Fluorescence intensity profile along the white dotted line in (**a**) is shown. c, Cells were stimulated with DMXAA for the indicated times. The fluorescence intensity of mRuby3-STING in cells was quantified. **d**, *Sting*[−/−] MEFs stably expressing mRuby3-STING and Lamp1-EGFP were treated with HT-DNA for 12 h. The cells were then treated with DMXAA and protease inhibitors for 12 h. The boxed area is magnified in the right panel. **e**, The ratio (%) of [mRuby3-STING inside or outside Lamp1[+]]/ [mRuby3-STING in whole cell] is indicated. **f**, Atg5 Tet-off cells were cultured with or without Doxycycline (Dox). Cells were stimulated with dsDNA for the indicated times. Cell lysates were analyzed by western blotting. **g**, mRuby3-STING and Lamp1-EGFP were stably expressed in Atg5 Tet-off cells. Cells were cultured with or without Dox for 4 days. The cells were then treated with DMXAA and protease inhibitors for 12 h. The boxed areas are magnified and shown. Fluorescence intensity profiles along the yellow lines are shown. **h**, The ratio (%) of [mRuby3-STING inside or outside Lamp1[+]]/[mRuby3-STING in whole cell] in (**g**) is indicated. Data are presented in box-and-whisker plots with the minimum, maximum, sample median, and first vs. third quartiles (**c**, **e**, and **h**). Scale bars, 10 μm in (**a**, **d**, and **g**), 500 nm in the magnified images in (**a**, **d**, and **g**). NS, not significant. The sample size (n) represents the number of cells (**c, e**, and **h**). Source numerical data and unprocessed blots are available in source data.

## a

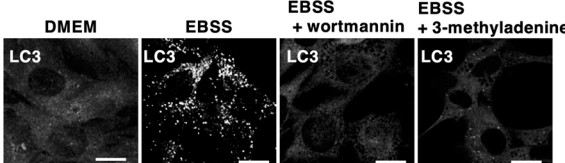

## b

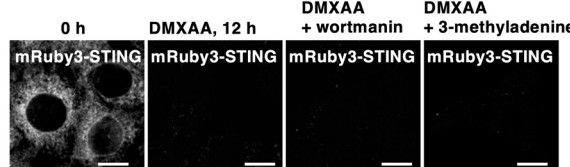

## c

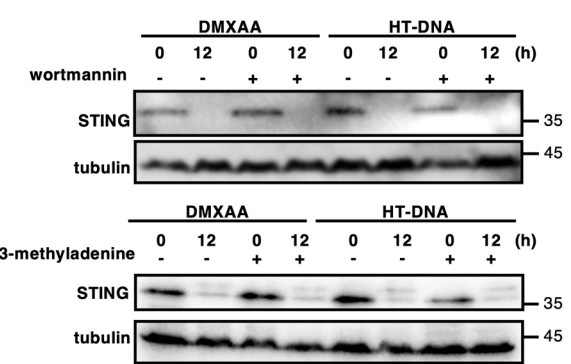

## d

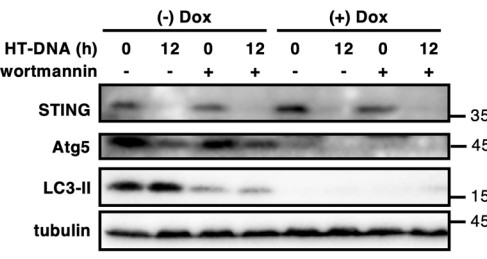

## e

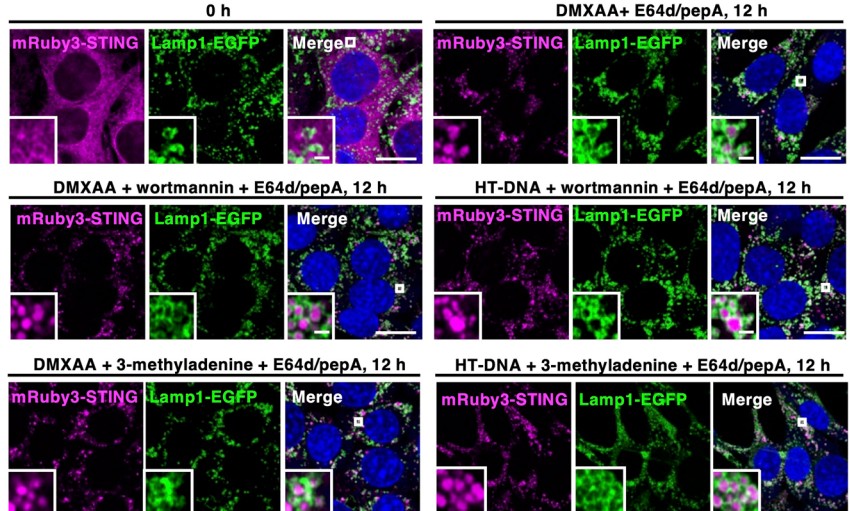

**Extended Data Fig. 2 | PI3K inhibitors did not inhibit STING degradation.**
**a**, MEFs were incubated with DMEM or EBSS for 2 h. For the inhibition of PI3K (phosphatidylinositol-3 kinase), wortmannin (1 µM) or 3-methyladenine (1 µM) was added to the medium. Cells were fixed, permeabilized, and immunostained with anti-LC3 antibody. **b**, *Sting*[−/−] MEFs stably expressing mRuby3-STING were treated with DMXAA (25 µg ml[−1]) for 0 or 12 h in the presence of wortmannin (1 µM) or 3-methyladenine (1 µM). Cells were fixed and imaged. **c**, MEFs were treated with DMXAA (25 µg ml[−1]) or HT-DNA (4 µg ml[−1]) for 12 h in the presence of wortmannin (1 µM) or 3-methyladenine (1 µM). Cell lysates were then prepared and analyzed by western blotting. **d**, Atg5 Tet-off cells were cultured with or without Doxycycline (Dox) (10 ng ml[−1]). Cells were stimulated with HT-DNA

(4 µg ml[−1]) for the indicated times in the presence of wortmannin (1 µM). Cell lysates were then prepared and analyzed by western blotting. **e**, *Sting*[−/−] MEFs stably expressing mRuby3-STING (magenta) and Lamp1-EGFP (green) were treated with DMXAA (25 µg ml[−1]) or HT-DNA (4 µg ml[−1]) for 0 or 12 h. For the inhibition of lysosomal proteolysis, E64d (30 µg ml[−1]) and pepstatin A (40 µg ml[−1]) were added to the medium. For the inhibition of PI3K, wortmannin (1 µM) or 3-methyladenine (1 µM) was added to the medium. Cells were fixed and imaged by Airyscan super-resolution microscopy. Scale bars, 10 µm in (**a**, **b**, and **e**), 500 nm in the magnified images in (**e**). Unprocessed blots are available in source data.

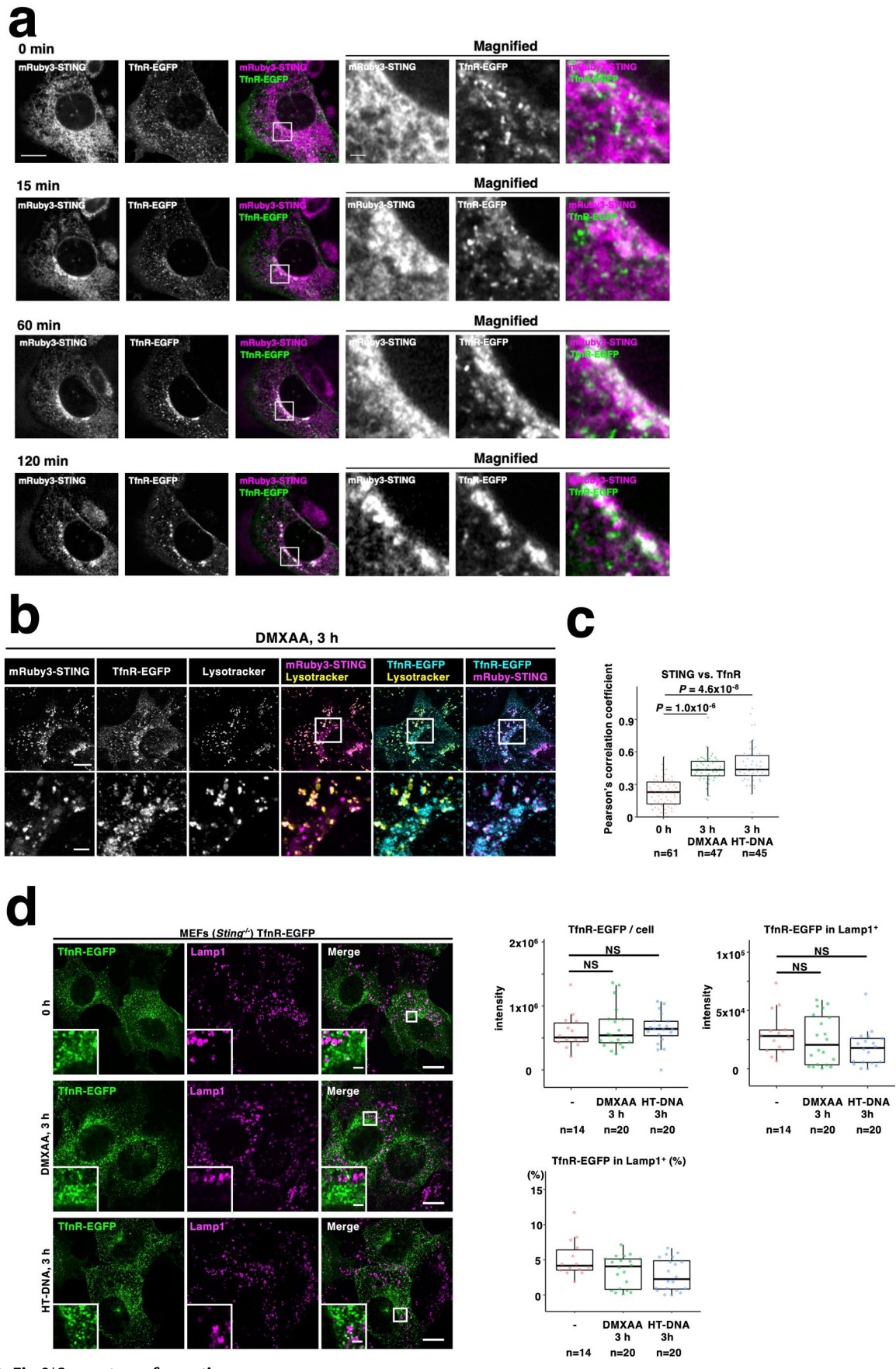

**Extended Data Fig. 3 | See next page for caption.**

**Extended Data Fig. 3 | STING co-localized with TfnR-EGFP after stimulation.**
**a**, *Sting*$^{-/-}$ MEFs stably expressing mRuby3-STING and TfnR-EGFP were imaged by Airyscan super-resolution microscopy every 1 min after stimulation with DMXAA. Selected images from the movie are shown. See also Supplementary video 1. The boxed areas in the merged images are magnified and shown. **b**, TfnR-EGFP (cyan) and mRuby3-STING (magenta) were stably expressed in *Sting*$^{-/-}$ MEFs. Cells were treated with DMXAA for 3 h and then with LysoTracker Deep Red (yellow). Live cell imaging was performed with Airyscan super-resolution microscopy. The boxed areas in the top panels are magnified in the bottom panels. **c**, The Pearson's correlation coefficient between mRuby3-STING and TfnR-EGFP is presented in box-and-whisker plots with the minimum, maximum, sample median, and first vs. third quartiles. **d**, *Sting*$^{-/-}$ MEFs stably expressing TfnR-EGFP were treated with DMXAA (25 μg ml$^{-1}$) or HT-DNA (4 μg ml$^{-1}$) for 3 h. Cells were fixed, permeabilized, and immunostained with anti-Lamp1 antibody. The fluorescence intensity of TfnR-EGFP in Lamp1-positive compartments (Lamp1$^+$) or in whole cell was quantified. Data are presented in box-and-whisker plots with the minimum, maximum, sample median, and first vs. third quartiles. Scale bars, 10 μm in (**a**, **b**, and **d**), 1 μm in the magnified images in (**a**, **b**, and **d**). The sample size (n) represents the number of cells (**c** and **d**). Source numerical data are available in source data.

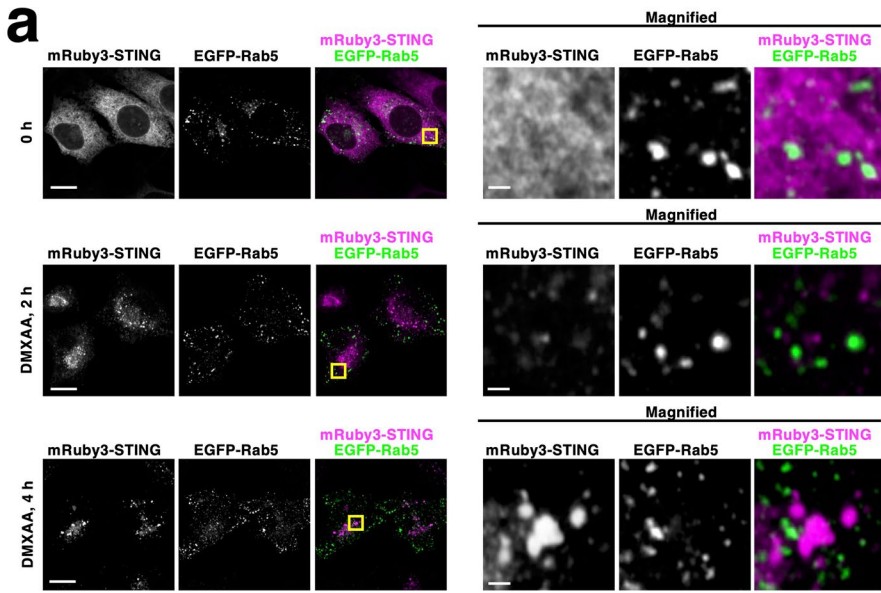

**Extended Data Fig. 4 | See next page for caption.**

**Extended Data Fig. 4 | STING was directly encapsulated by Lamp1⁺ compartments. a**, EGFP-Rab5a (green) and mRuby3-STING (magenta) were stably expressed *Sting*⁻/⁻ MEFs. Cells were treated with DMXAA for the indicated times. Cells were fixed and imaged by Airyscan super-resolution microscopy. The yellow boxed areas are magnified and shown in the right panels. **b**, *Sting*⁻/⁻ MEFs stably expressing mRuby3-STING were treated with DMXAA ($25\ \mu g\ ml^{-1}$) as indicated. Cells were then fixed, permeabilized, and immunostained with anti-EEA1 or anti-LBPA antibody. Data are presented in box-and-whisker plots with the minimum, maximum, sample median, and first vs. third quartiles as the ratio (%) of [mRuby3-STING inside endosome]/[mRuby3-STING in whole cell]. The white boxes are magnified and shown in the bottom panels. **c-e**, Cells were imaged by Airyscan super-resolution microscopy every 5 seconds from 3 hours after DMXAA stimulation. The time-lapse images of the regions outlined by the yellow boxes in (**c**) are shown sequentially in (**d**) and (**e**). The dotted green lines indicate the limiting membrane of lysosome. Scale bars 10 μm in (**a**, **b**, and **c**), 500 nm in (**d** and **e**), 500 nm in the magnified images in (**a** and **b**). The sample size (n) represents the number of cells (**b**). Source numerical data are available in source data.

## a   related to Figure 3b

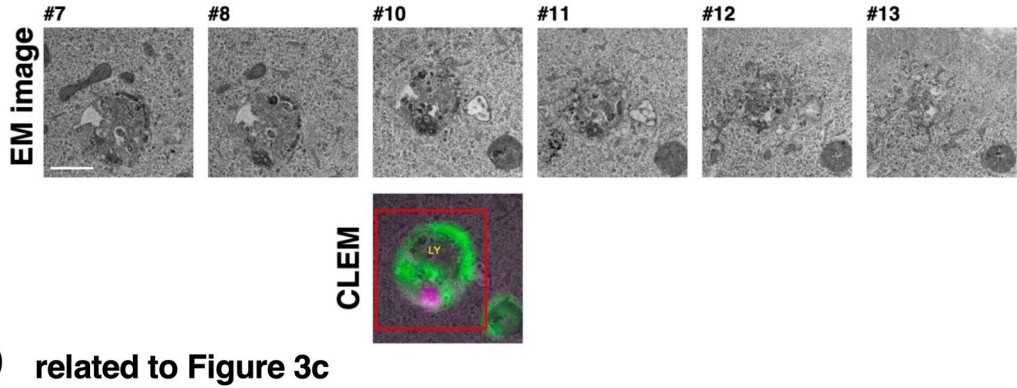

## b   related to Figure 3c

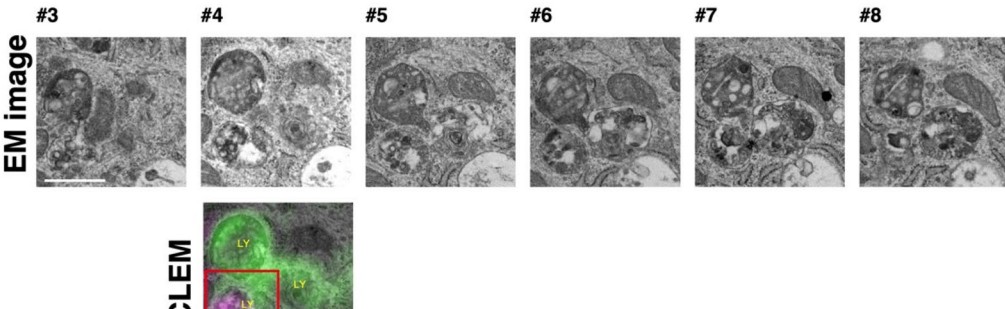

## c   related to Figure 3d

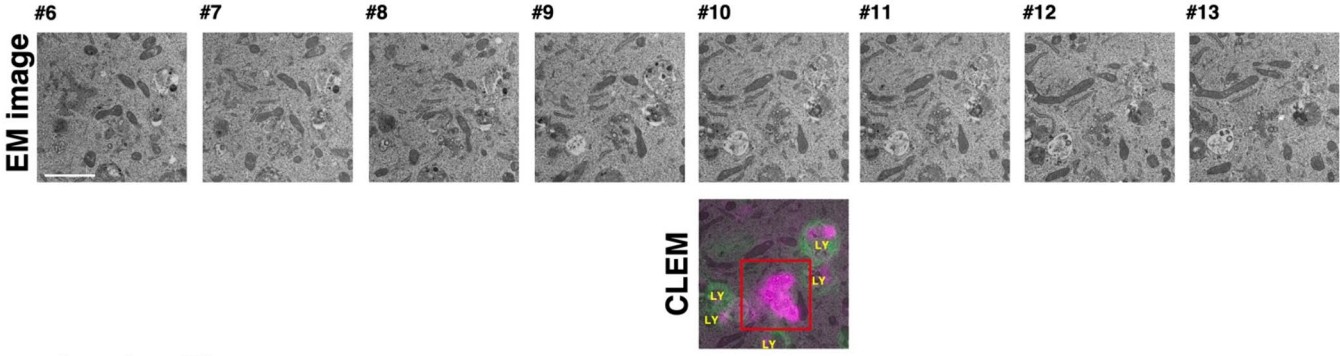

## d   related to Figure 3e

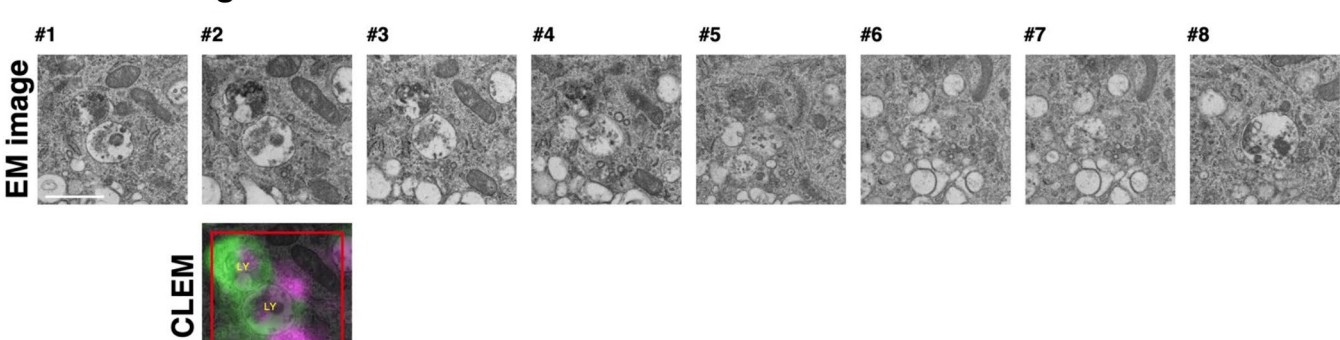

**Extended Data Fig. 5 | CLEM analysis of cluster of STING vesicles.** Serial EM pictures and a selected CLEM image are shown. Panels **a**, **b**, **c**, and **d** correspond to Fig. 3b–e, respectively. The number (#) indicates the order in the serial section. LY, Lysosomes; Scale bars, 500 nm.

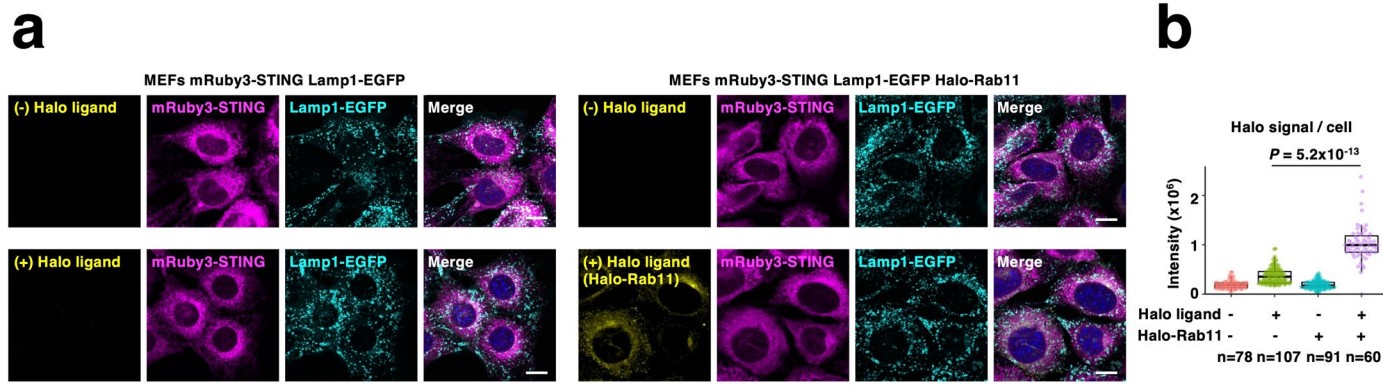

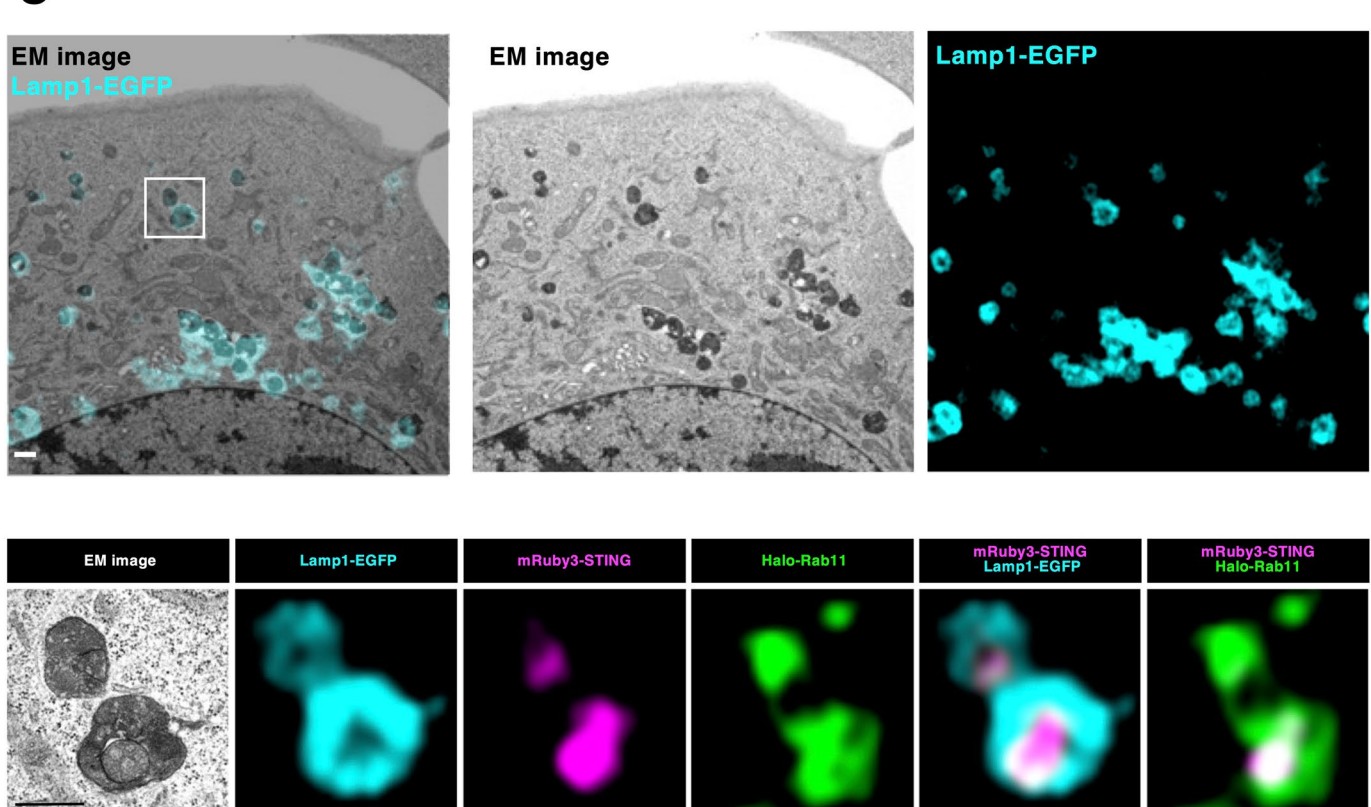

**Extended Data Fig. 6 | STING co-localized with Rab11 within lysosomes.**
**a**, *Sting$^{-/-}$* MEFs stably expressing mRuby3-STING (magenta), Lamp1-EGFP (cyan) were treated with or without HaloTag SaraFluor 650 T ligand (yellow) (the left 4 panels). *Sting$^{-/-}$* MEFs stably expressing mRuby3-STING (magenta), Lamp1-EGFP (cyan), and Halo-Rab11a (yellow) were treated with or without HaloTag SaraFluor 650 T ligand (the right 4 panels). Scale bars, 10 μm. **b**, The far-red fluorescence intensity of the cells was quantified. Data are presented in box-and-whisker plots with the minimum, maximum, sample median, and first vs. third quartiles. **c**, *Sting$^{-/-}$* MEFs stably expressing mRuby3-STING (magenta), Lamp1-EGFP (cyan), and Halo-Rab11a (green) were treated with DMXAA for 6 h in the presence of E64d/pepstatin A/orlistat. Scale bars, 500 nm. The sample size (n) represents the number of cells (**b**). Source numerical data are available in source data.

## a

| | Mammal Vps Genes | Yeast vps genes | Suppression of STING degradation |
|---|---|---|---|
| - | Atp6v1b2 | | 2.711 |
| 1 | Stx7 | vps6 | 2.089 |
| 2 | Chmp2a | vps2 (Chmp2) | 1.947 |
| 3 | Slc9a9 | vps44 | 1.673 |
| 4 | Slc2a3 | vps73 | 1.590 |
| 5 | Rab5C | vps2 (Rab5) | 1.548 |
| 6 | Vps39 | vps39 | 1.473 |
| 7 | Vps36 | vps36 | 1.470 |
| 8 | Tmem50b | vps68 | 1.411 |
| 9 | Vps37a | vps37a | 1.411 |
| 10 | Snx32 | vps17 (Snx32) | 1.386 |
| 11 | Rbsn | vps19 | 1.364 |
| 12 | Sorcs2 | vps10 (Sorcs2) | 1.314 |
| 13 | Chmp4b | vps32 (Chmp4) | 1.300 |
| 14 | Tsg101 | vps23 | 1.299 |
| 15 | Rabgef1 | vps9 | 1.289 |
| 16 | Ndst2 | vps65 | 1.282 |
| 17 | Rab5b | vps21 (Rab5) | 1.261 |
| 18 | Golph3 | vps74 | 1.253 |
| 19 | Naalad2 | vps70 | 1.243 |
| 20 | Vps53 | vps53 | 1.235 |
| 21 | Vps4b | vps4b | 1.229 |
| 22 | Pik3r4 | vps15 | 1.227 |
| 23 | Vps28 | vps28 | 1.212 |
| 24 | Vps52 | vps52 | 1.204 |
| 25 | Leprot | vps55 | 1.198 |
| 26 | Vps37b | vps37b | 1.178 |
| 27 | Chmp5 | vps60 | 1.169 |
| 28 | Vps25 | vps25 | 1.169 |
| 29 | Snx1 | vps5 (Snx1) | 1.167 |
| 30 | Vps45 | vps45 | 1.164 |
| 31 | PDCD6IP | vps31 | 1.164 |
| 32 | Vps37c | vps37c | 1.157 |
| 33 | Pik3c3 | vps34 | 1.121 |
| 34 | Tspyl5 | vps75 | 1.105 |
| 35 | Sorcs3 | vps10 (Sorcs3) | 1.104 |
| 36 | Rab5A | vps21 (Rab5) | 1.091 |
| 37 | Chmp2b | vps2 (Chmp2) | 1.082 |
| 38 | Sorl1 | vps10 (Sorl1) | 1.080 |
| 39 | Vps72 | vps72 | 1.075 |
| 40 | Chmp4C | vps32 (Chmp4) | 1.072 |
| 41 | Vps38 | vps38 | 1.069 |
| 42 | Vps37d | vps37d | 1.069 |
| 43 | Chmp3 | vps24 | 1.068 |
| 44 | HGS | vps27 | 1.065 |
| 45 | Snx2 | vps5 (Snx2) | 1.053 |
| 46 | Becn1 | vps30 | 1.049 |
| 47 | Vps8 | vps8 | 1.045 |
| 48 | Chmp1a | vps46 | 1.038 |
| 49 | Vps26a | vps26a | 1.035 |
| 50 | Snx5 | vps17 (Snx5) | 1.020 |
| 51 | Vps35 | vps35 | 1.019 |
| 52 | Enkur | vps63 (Enkur) | 1.004 |
| 53 | Vps54 | vps54 | 1.004 |
| 54 | Sorcs1 | vps10 (Sorcs1) | 1.003 |
| 55 | Vps16 | vps16 | 1.000 |
| 56 | control | | 1.000 |
| 57 | Dscr3 Vps26c | vps26c | 0.970 |
| 58 | Vps4a | vps4a | 0.966 |
| 59 | Vps51 | vps51 | 0.965 |
| 60 | Mical1 | vps63 (Mical1) | 0.960 |
| 61 | Vps41 | vps41 | 0.951 |
| 62 | Chmp6 | vps20 | 0.950 |
| 63 | Vps33a | vps33a | 0.938 |
| 64 | Snf8 | vps22 | 0.929 |
| 65 | Vps18 | vps18 | 0.928 |
| 66 | Vps29 | vps29 | 0.909 |
| 67 | Snx6 | vps17 (Snx6) | 0.908 |
| 68 | DNM1L | vps1 | 0.907 |
| 69 | Vps50 | vps50 | 0.896 |
| 70 | Vps11 | vps11 | 0.891 |
| 71 | Vps13b | vps13b | 0.869 |
| 72 | Vps13c | vps13c | 0.855 |
| 73 | Vps13a | vps13a | 0.854 |
| 74 | Vps33b | vps33b | 0.828 |
| 75 | Sort1 | vps10 (Sort1) | 0.793 |
| 76 | Vps26b | vps26b | 0.743 |

## b

| | Mammal Vps Genes | Yeast vps genes | STING-dependent type-I IFN |
|---|---|---|---|
| 1 | Vps26a | vps26a | 3.120 |
| 2 | Vps28 | vps28 | 2.201 |
| 3 | Tsg101 | vps23 | 2.062 |
| 4 | Vps39 | vps39 | 2.005 |
| 5 | Vps50 | vps50 | 1.926 |
| 6 | Snx2 | vps5 (Snx2) | 1.882 |
| 7 | Vps11 | vps11 | 1.848 |
| 8 | Vps29 | vps29 | 1.837 |
| 9 | Vps33b | vps33b | 1.819 |
| 10 | Vps37b | vps37b | 1.758 |
| 11 | Vps45 | vps45 | 1.754 |
| 12 | Vps18 | vps18 | 1.717 |
| 13 | Vps16 | vps16 | 1.701 |
| 14 | Vps51 | vps51 | 1.697 |
| 15 | Vps13b | vps13b | 1.681 |
| 16 | Vps41 | vps41 | 1.666 |
| 17 | Vps13c | vps13c | 1.635 |
| 18 | Vps37a | vps37a | 1.633 |
| 19 | Vps33a | vps33a | 1.572 |
| 20 | Chmp4b | vps32 (Chmp4) | 1.518 |
| 21 | Enkur | vps63 (Enkur) | 1.433 |
| 22 | Vps13a | vps13a | 1.417 |
| 23 | Pik3r4 | vps15 | 1.352 |
| 24 | Vps4b | vps4b | 1.349 |
| 25 | Snx5 | vps17 (Snx5) | 1.331 |
| 26 | DNM1L | vps1 | 1.317 |
| 27 | Vps25 | vps25 | 1.240 |
| 28 | Vps38 | vps38 | 1.228 |
| 29 | Vps37d | vps37d | 1.228 |
| 30 | Chmp6 | vps20 | 1.207 |
| 31 | Chmp1a | vps46 | 1.198 |
| 32 | Slc2a3 | vps73 | 1.153 |
| 33 | Vps36 | vps36 | 1.145 |
| 34 | Ndst2 | vps65 | 1.129 |
| 35 | Mical1 | vps63 (Mical1) | 1.119 |
| 36 | Sorcs1 | vps10 (Sorcs1) | 1.077 |
| 37 | Rab5c | vps21 (Rab5) | 1.067 |
| 38 | Pik3c3 | vps34 | 1.061 |
| 39 | Chmp3 | vps24 | 1.042 |
| 40 | Tmem50b | vps68 | 1.002 |
| 41 | control | | 1.000 |
| 42 | Vps35 | vps35 | 0.992 |
| 43 | Vps72 | vps72 | 0.960 |
| 44 | Snx6 | vps17 (Snx6) | 0.951 |
| 45 | Rab5A | vps21 (Rab5a) | 0.945 |
| 46 | Vps53 | vps53 | 0.914 |
| 47 | Vps26b | vps26b | 0.879 |
| 48 | Sort1 | vps10 (Sort1) | 0.877 |
| 49 | Snx32 | vps17 (Snx32) | 0.866 |
| 50 | PDCD6IP | vps31 | 0.835 |
| 51 | Dscr3 Vps26c | vps26c | 0.770 |
| 52 | Chmp2a | vps2 (Chmp2) | 0.763 |
| 53 | Tspyl5 | vps75 | 0.759 |
| 54 | Chmp2b | vps2 (Chmp2) | 0.720 |
| 55 | Naalad2 | vps70 | 0.719 |
| 56 | Rab5b | vps21 (Rab5) | 0.715 |
| 57 | Vps52 | vps52 | 0.666 |
| 58 | Slc9a9 | vps44 | 0.621 |
| 59 | Golph3 | vps74 | 0.619 |
| 60 | Rbsn | vps19 | 0.607 |
| 61 | Sorl1 | vps10 (Sorl1) | 0.593 |
| 62 | Vps54 | vps54 | 0.565 |
| 63 | Sorcs3 | vps10 (Sorcs3) | 0.564 |
| 64 | HGS | vps27 | 0.546 |
| 65 | Sorcs2 | vps10 (Sorcs2) | 0.541 |
| 66 | Becn1 | vps30 | 0.534 |
| 67 | Vps8 | vps8 | 0.523 |
| 68 | Leprot | vps55 | 0.509 |
| 69 | Stx7 | vps6 | 0.466 |
| 70 | Snf8 | vps22 | 0.385 |
| 71 | Vps4a | vps4a | 0.352 |
| 72 | Rabgef1 | vps9 | 0.348 |
| 73 | Chmp4C | vps32 (Chmp4c) | 0.335 |
| 74 | Chmp5 | vps60 | 0.317 |
| 75 | Snx1 | vps5 (Snx1) | 0.153 |
| 76 | Vps37c | vps37c | 0.151 |
| - | Atp6V1b2 | | 0.060 |

## c

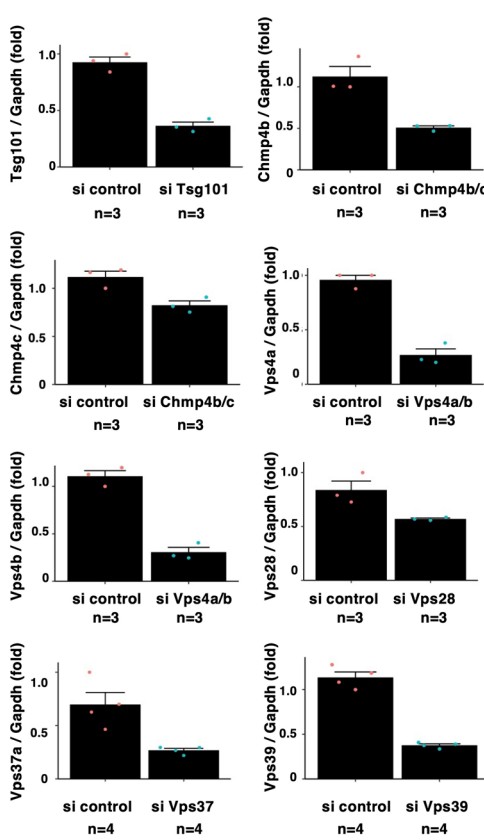

■ Vps genes ranked within top 25 in Fig. 4b and c (red)
■ Vps genes ranked within top 25 in Fig. 4b (pink)
■ Vps genes ranked within top 25 in Fig. 4b and c (blue)
■ Vps genes ranked within top 25 in Fig. 4c (light blue)

**Extended Data Fig. 7 | See next page for caption.**

**Extended Data Fig. 7 | Vps genes involved in STING degradation and the termination of type I interferon response. a**, The data related to Fig. 4b are shown. The top 25 genes are highlighted in red. The genes that were also ranked within top 25 in 'type I interferon assay' are highlighted in bright red. **b**, The data related to Fig. 4c are shown. The top 25 genes are highlighted in blue. The genes that were also ranked within top 25 in 'STING degradation assay' are highlighted in bright blue. **c**, Knockdown efficiency of Vps genes. Cells were treated with the indicated siRNAs for 72 hours, and qRT-PCR was performed. Gapdh was used as an internal control. Data are presented as mean values +/− SEM. The sample size (n) represents the number of the biological replicates (**c**). Source numerical data are available in source data.

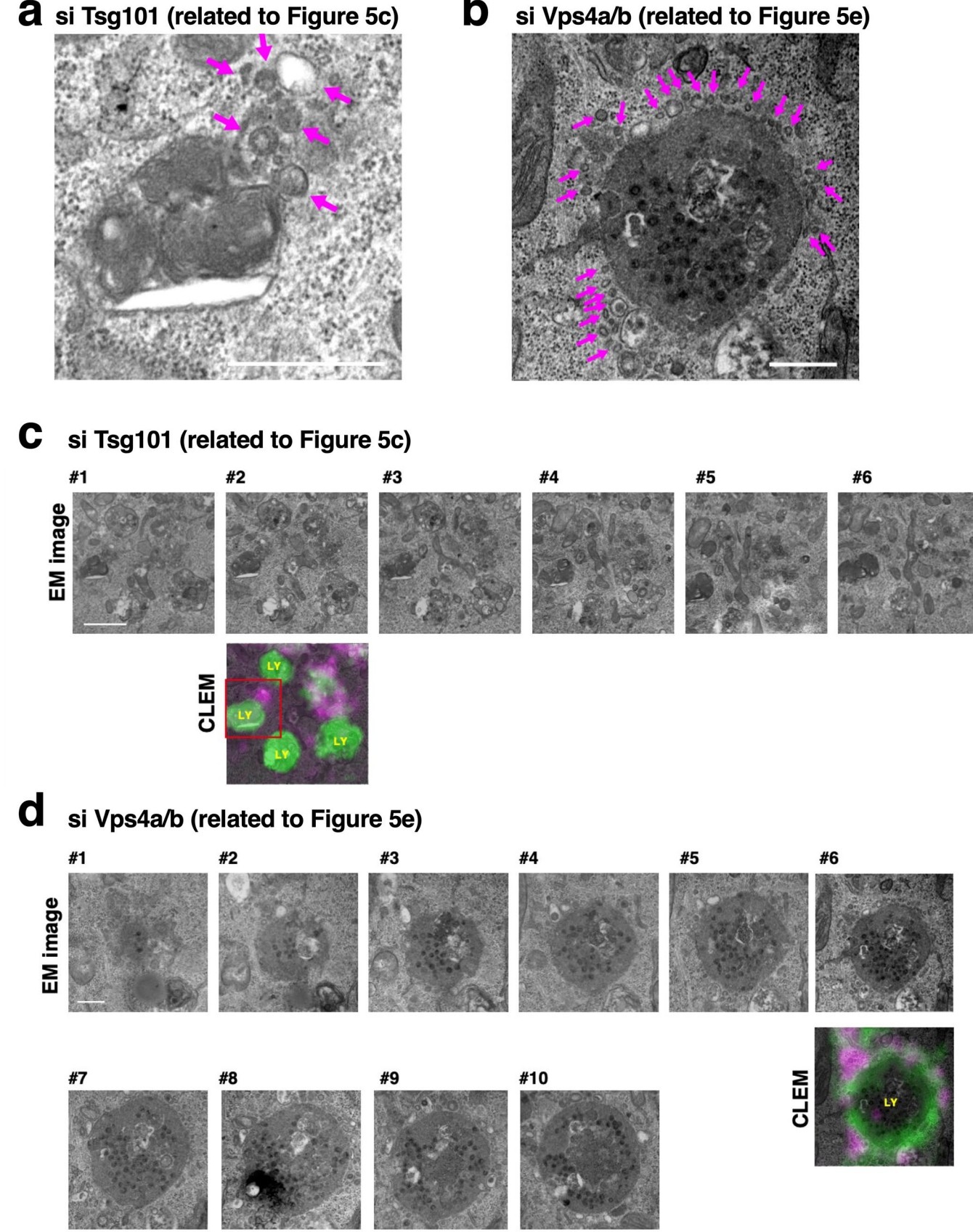

**Extended Data Fig. 8 | CLEM analyses of Tsg101- or Vps4a/b-depleted cells. a**, The magnified EM picture of Fig. 5c. **b**, The magnified EM picture of Fig. 5e. **c** and **d**, Serial EM pictures and a selected CLEM image are shown. Panels (**c**) and (**d**) correspond to Fig. 5c, e, respectively. The number (#) indicates the order in the serial section. LY, Lysosomes; Scale bars, 500 nm.

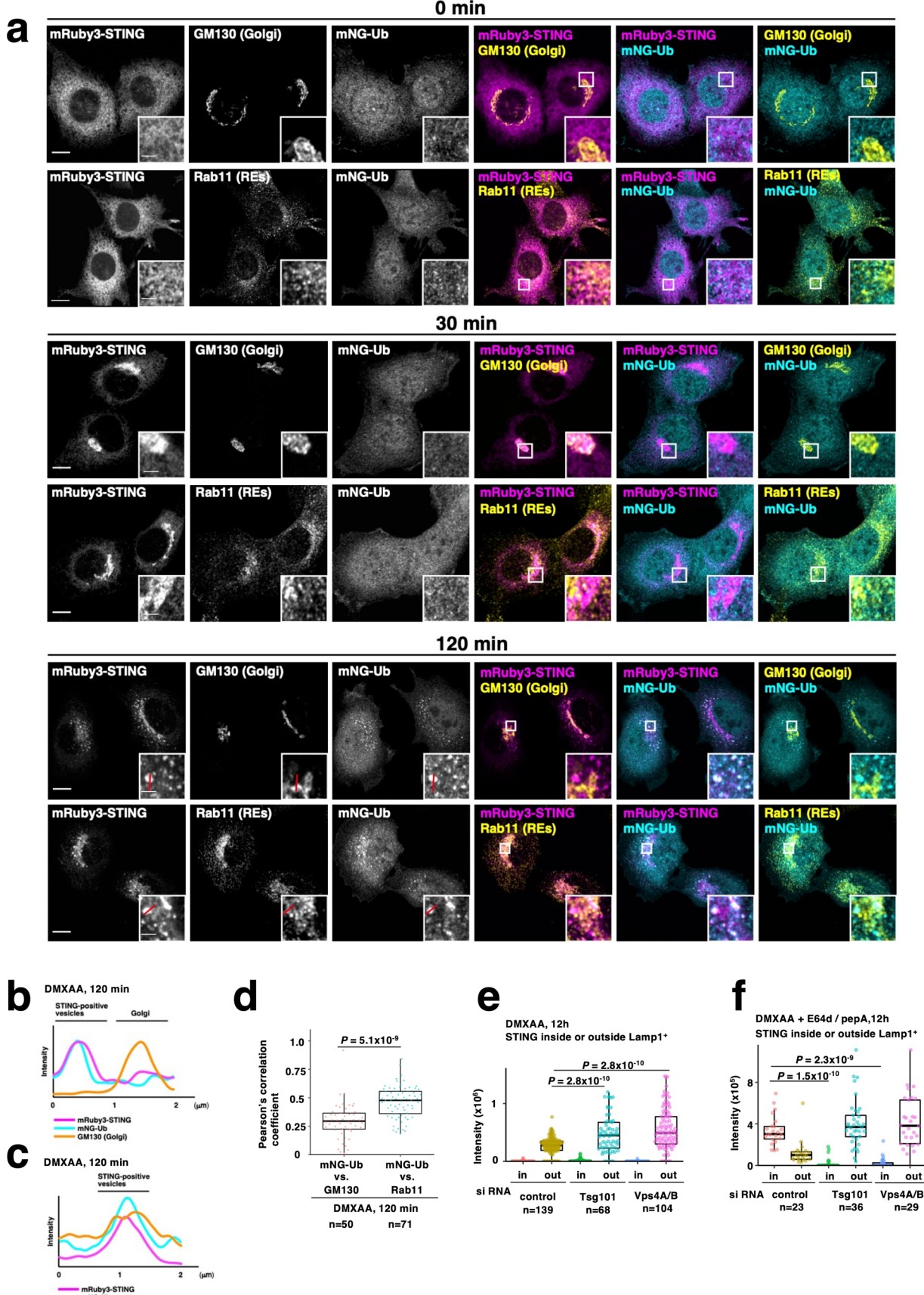

**Extended Data Fig. 9 | See next page for caption.**

**Extended Data Fig. 9 | STING co-localized with mNeonGreen-ubiquitin at Rab11-positive REs after stimulation. a**, mNeonGreen-ubiquitin (mNG-Ub, cyan) and mRuby3-STING (magenta) were stably expressed in *Sting*[-/-] MEFs. Cells were stimulated with DMXAA for the indicated times. Cells were then fixed, permeabilized, and immunostained with anti-GM130 (a Golgi protein, yellow) or anti-Rab11 (a recycling endosomal protein, yellow) antibodies. Scale bars, 10 μm, 1 μm in the magnified images. **b**, **c**, Fluorescence intensity profiles along the red lines in (**a**) are shown. **d**, The Pearson's correlation coefficient between mNeonGreen-ubiquitin and GM130, or between mNeonGreen-ubiquitin and Rab11 are presented in box-and-whisker plots with the minimum, maximum, sample median, and first vs. third quartiles. **e**, *Sting*[-/-] MEFs reconstituted with mRuby3-STING were treated with indicated siRNAs. Cells were then incubated with DMXAA for 12 h. The fluorescence intensity of mRuby3-STING inside or outside Lamp1-positive compartments (Lamp1[+]) was quantified. Data are presented in box-and-whisker plots with the minimum, maximum, sample median, and first vs. third quartiles. **f**, *Sting*[-/-] MEFs stably expressing mRuby3-STING and Lamp1-EGFP were treated with indicated siRNAs. Cells were then incubated with DMXAA and E64d/pepstatin A for 12 h. The fluorescence intensity of mRuby3-STING inside or outside Lamp1-positive compartments (Lamp1[+]) was quantified. Data are presented in box-and-whisker plots with the minimum, maximum, sample median, and first vs. third quartiles. The sample size (n) represents the number of cells (**d**, **e**, and **f**). Source numerical data are available in source data.

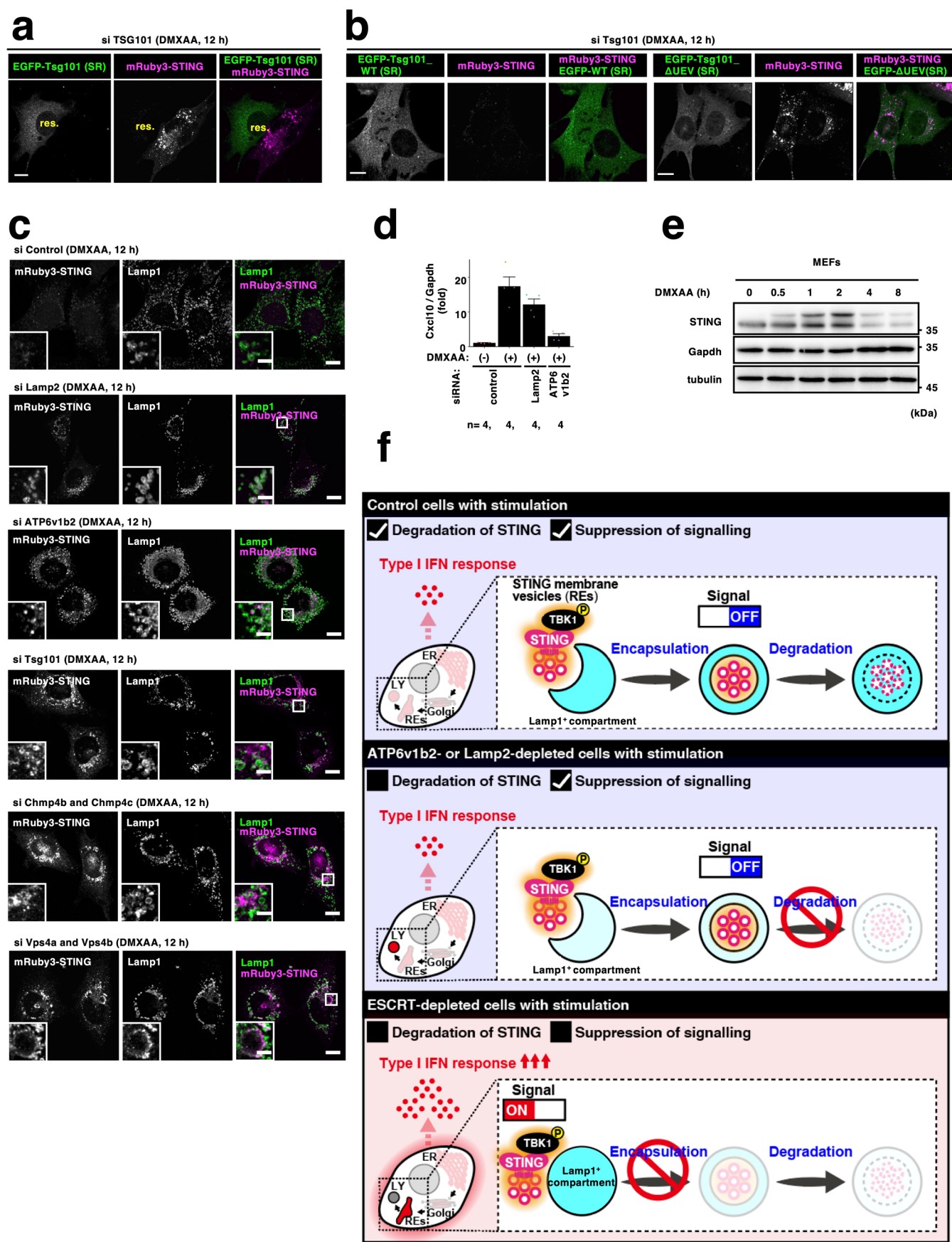

**Extended Data Fig. 10 | See next page for caption.**

**Extended Data Fig. 10 | ESCRT-driven microautophagy terminates STING signaling. a**, Two cell lines [*Sting*⁻/⁻ MEFs expressing mRuby3-STING, and *Sting*⁻/⁻ MEFs expressing mRuby3-STING and siRNA resistant EGFP-Tsg101 (WT)] were mixed, treated with Tsg101 siRNA, and stimulated with DMXAA for 12 h. **b**, siRNA-resistant EGFP-Tsg101 (WT or ΔUEV) and mRuby3-STING were stably expressed in *Sting*⁻/⁻ MEFs. Cells were treated with Tsg101 siRNA, and stimulated with DMXAA for 12 h. **c**, *Sting*⁻/⁻ MEFs expressing mRuby3-STING were treated with the indicated siRNAs and stimulated with DMXAA for 12 h. Cells were then fixed, permeabilized, and immunostained with anti-Lamp1. Scale bars, 10 μm, 500 nm in the magnified images. **d**, qRT-PCR of the expression of Cxcl10 in MEFs that were treated with the indicated siRNAs followed by stimulation with DMXAA for 12 h. Data are presented as mean values +/− SD. **e**, Western blots of cell lysates of MEFs stimulated with DMXAA for the indicated times. The sample size (n) represents the number of the biological replicates (**d**). Source numerical data and unprocessed blots are available in source data. **f**, (**Control cells**) Active STING/TBK1 complex is encapsulated into the lumen of Lamp1-positive compartments by microautophgy. Thus the signalling is terminated. (**ATP6v1b2- or Lamp2-depleted cells**) The encapsulation of active STING/TBK1 complex proceeds (Extended Data Fig. 10c). Thus the signalling is terminated (Extended Data Fig. 10d). Please be noted that STING degradation is impaired because of the defect in the ability of lysosomal proteolysis. (**ESCRT-depleted cells**) The encapsulation is impaired, thus active STING/TBK1 complex remains in the cytosol, leading to the duration of the signalling. Recent studies indicate the involvement of C9orf72 (PMID: 32814898), BLOC1(PMID: 29033128), NPC1(PMID: 34290407) in the STING degradation. The exact site of action of these proteins in the membrane traffic that STING follows, namely, 'ER-the Golgi-REs-lysosomes' remains to be elucidated.

# Reporting Summary

## Statistics

For all statistical analyses, confirm that the following items are present in the figure legend, table legend, main text, or Methods section.

| n/a | Confirmed | |
|---|---|---|
| ☐ | ☒ | The exact sample size (*n*) for each experimental group/condition, given as a discrete number and unit of measurement |
| ☐ | ☒ | A statement on whether measurements were taken from distinct samples or whether the same sample was measured repeatedly |
| ☐ | ☒ | The statistical test(s) used AND whether they are one- or two-sided<br>*Only common tests should be described solely by name; describe more complex techniques in the Methods section.* |
| ☐ | ☒ | A description of all covariates tested |
| ☐ | ☒ | A description of any assumptions or corrections, such as tests of normality and adjustment for multiple comparisons |
| ☐ | ☒ | A full description of the statistical parameters including central tendency (e.g. means) or other basic estimates (e.g. regression coefficient) AND variation (e.g. standard deviation) or associated estimates of uncertainty (e.g. confidence intervals) |
| ☐ | ☒ | For null hypothesis testing, the test statistic (e.g. *F*, *t*, *r*) with confidence intervals, effect sizes, degrees of freedom and *P* value noted<br>*Give P values as exact values whenever suitable.* |
| ☒ | ☐ | For Bayesian analysis, information on the choice of priors and Markov chain Monte Carlo settings |
| ☒ | ☐ | For hierarchical and complex designs, identification of the appropriate level for tests and full reporting of outcomes |
| ☐ | ☒ | Estimates of effect sizes (e.g. Cohen's *d*, Pearson's *r*), indicating how they were calculated |

*Our web collection on statistics for biologists contains articles on many of the points above.*

## Software and code

Policy information about availability of computer code

| | |
|---|---|
| Data collection | Western blot data were collected using FUSION SOLO (software;Evolution Capt)<br>Microscopy data were collected using Zeiss ZEN 2.3 SP1 FP3 (black, 64 bit) (ver. 14.0.21.201)<br>CLEM data were collected sing JEM1400EX; JEOL.<br>The data of the luciferase activity was collected  by GloMax® Navigator Microplate Luminometer (Promega) (version 3.1.0).<br>Quantitative real-time PCR (qRT-PCR) was performed using LightCycler 96 (Roche). |
| Data analysis | Western blot data were analysed by Fiji (ver. 2.1.0/1.53c).<br>Microscopy data  were analysed by Fiji (ver. 2.1.0/1.53c) including the Trainable Weka Segmentation plugin (v3.3.2),  Cellpose (v1.0), R (ver. 4.1.2), and KNIME (ver. 4.5.1).<br>Proteomics data were analysed by MASCOT (ver. 2.6), Proteome Discover (ver. 2.2). |

For manuscripts utilizing custom algorithms or software that are central to the research but not yet described in published literature, software must be made available to editors and reviewers. We strongly encourage code deposition in a community repository (e.g. GitHub). See the Nature Portfolio guidelines for submitting code & software for further information.

# Data

Policy information about availability of data

All manuscripts must include a data availability statement. This statement should provide the following information, where applicable:

- Accession codes, unique identifiers, or web links for publicly available datasets
- A description of any restrictions on data availability
- For clinical datasets or third party data, please ensure that the statement adheres to our policy

The authors declare that the data supporting the findings of this study are available within the supplementary information. NCBI nr database is available in (https://www.ncbi.nlm.nih.gov). Mass spectrometry data have been deposited in ProteomeXchange with the primary accession code PXD039411.

# Field-specific reporting

Please select the one below that is the best fit for your research. If you are not sure, read the appropriate sections before making your selection.

☒ Life sciences ☐ Behavioural & social sciences ☐ Ecological, evolutionary & environmental sciences

For a reference copy of the document with all sections, see nature.com/documents/nr-reporting-summary-flat.pdf

# Life sciences study design

All studies must disclose on these points even when the disclosure is negative.

| | |
|---|---|
| Sample size | No sample size calculation was applied in this study to predetermine sample sizes for experiments using cell lines. A sample size of three or more was used as to evaluate the spread of the data and was determined based upon other studies with similar methodologies (PMID: 27324217, 29093443, 33397928) |
| Data exclusions | No data have been excluded from any analysis. |
| Replication | All experiments have been repeated at least three times independently, and each yielding similar results. |
| Randomization | Randomization was not relevant for cell culture study, because all cells used in this study had to be differently treated and analyzed in parallel to minimize experimental variation. |
| Blinding | All the experiments were unblinded because these experiments were not susceptible to bias. |

# Reporting for specific materials, systems and methods

We require information from authors about some types of materials, experimental systems and methods used in many studies. Here, indicate whether each material, system or method listed is relevant to your study. If you are not sure if a list item applies to your research, read the appropriate section before selecting a response.

## Materials & experimental systems

| n/a | Involved in the study |
|---|---|
| ☐ | ☒ Antibodies |
| ☐ | ☒ Eukaryotic cell lines |
| ☒ | ☐ Palaeontology and archaeology |
| ☐ | ☒ Animals and other organisms |
| ☐ | ☒ Human research participants |
| ☒ | ☐ Clinical data |
| ☒ | ☐ Dual use research of concern |

## Methods

| n/a | Involved in the study |
|---|---|
| ☒ | ☐ ChIP-seq |
| ☐ | ☒ Flow cytometry |
| ☒ | ☐ MRI-based neuroimaging |

# Antibodies

| | |
|---|---|
| Antibodies used | Antibodies used in this study were as follows: anti-Atg5 (MBL Life science, PM050, dilution 1/1000, WB), anti-STING (proteintech, 19851-1-AP, dilution 1/1000, WB), anti-tubulin (Sigma-Aldrich, DM1A, dilution 1/1000, WB), anti-GM130 (BD Biosciences, 610823, dilution 1/1000, IF), anti-Rab11 (Cell Signaling Technology, D4F5, dilution 1/100, IF), anti-GAPDH (MERCK, 6C5, dilution 1/1000, WB), anti-pTBK1 (Cell Signaling Technology, D52C2, dilution 1/1000, IF/WB), anti-TBK1 (Abcam, ab40676, dilution 1/1000, WB), anti-ubiquitin(P4D1) (Abcam, ab139101, dilution 1/1000, WB), anti-K63 ubiquitin (millipore, 05-1308, dilution 1/100, IF), anti-GFP (Clontech, JL-8, dilution 1/1000, WB), anti-GFP (Thermo Fisher Scientific, 3E6, dilution 1/500, IP), anti-Goat Anti-Rabbit IgG (H+L) Mouse/Human ads-HRP (Southern Biotech, 4050-05, dilution 1/1000, WB), anti-Goat Anti-Mouse IgG (H+L) Human ads-HRP (Southern Biotech, 1031-05, dilution 1/1000, WB), anti-Alexa 488-, 594-, or 647- conjugated secondary antibodies (Thermo Fisher Scientific, A21202, A21203, A21206, A21207, A31573, A11016, A21448, dilution 1/1000, IF). anti-LC3 (MBL Life science, PM036 |

1/1000, IF/WB), anti-EEA1 (BD Biosciences, 610456, 1/500 IF), anti-LBPA (Merck Millipore, 6C4, 1/500, IF), anti-Clathrin heavy chain (Cell Signaling Technology, D3C6 ,1/1000 IF), anti-Tsg101 (Abcam, ab125011, 1/500, WB), anti-Lamp1 (eBioscience, 1D4B, 1/1000, IF)

| Validation | All antibodies were validated by the vendors and documented with corresponding data sheets as follows. |
|---|---|
| | mouse anti-Atg5 (MBL Life science, PM050): validated for mouse Atg5 by WB with cell lysate. |
| | rabbit anti-STING (proteintech, 19851-1-AP): validated for mouse STING by WB with cell lysate. |
| | mouse anti-tubulin (Sigma-Aldrich, DM1A): validated for mouse tubulin by WB with cell lysate. |
| | mouse anti-GM130 (BD Biosciences, 610823): validated for mouse GM130 by IF with fixed cells. |
| | rabbit anti-Rab11 (Cell Signaling Technology, D4F5): validated for mouse Rab11 by IF with fixed cells. |
| | rabbit anti-GAPDH (MERCK, 6C5): validated for mouse GAPDH by WB with cell lysate. |
| | rabbit anti-pTBK1 (Cell Signaling Technology, D52C2): validated for mouse pTBK1 by IF with fixed cells and by WB with cell lysate. |
| | rabbit anti-TBK1 (Abcam, ab40676): validated for mouse TBK1 by WB with cell lysate. |
| | mouse anti-ubiquitin(P4D1) (Abcam, ab139101): validated for mouse ubiquitin(P4D1) by WB with cell lysate. |
| | rabbit anti-K63 ubiquitin (millipore, 05-1308): validated for mouse K63 ubiquitin by IF with fixed cells. |
| | mouse anti-GFP (Clontech, JL-8): validated for mouse GFP by WB with cell lysate. |
| | mouse anti-GFP (Thermo Fisher Scientific, 3E6): validated for mouse GFP by IP with western blotting. |
| | Goat Anti-Rabbit IgG (H+L) Mouse/Human ads-HRP (Southern Biotech, 4050-05): validated for mouse Goat Anti-Rabbit IgG (H+L) Mouse/Human ads-HRP by WB with cell lysate. |
| | Goat Anti-Mouse IgG (H+L) Human ads-HRP (Southern Biotech, 1031-05): validated for mouse Goat Anti-Mouse IgG (H+L) Human ads-HRP by WB with cell lysate. |
| | donkey Alexa 488-, 594-, or 647- conjugated secondary antibodies (Thermo Fisher Scientific, A21202, A21203, A21206, A21207, A31573, A11016, A21448): validated for mouse Alexa 488-, 594-, or 647- conjugated secondary antibodies by IF with fixed cells. |
| | rabbit anti-LC3 (MBL Life science, PM036): validated for mouse LC3 by WB and IF with fixed cells. |
| | mouse anti-EEA1 (BD Biosciences, 610456): validated for mouse EEA1 by IF with fixed cells. |
| | mouse anti-LBPA (Merck Millipore, 6C4): validated for mouse LBPA by IF with fixed cells. |
| | rabbit anti-Clathrin heavy chain (Cell Signaling Technology, D3C6) : validated for mouse Clathrin heavy chain by IF with fixed cells. |
| | rabbit anti-Tsg101 (Abcam, ab125011, 1/500, WB): validated for mouse/human Tsg101 by WB. |
| | mouse anti-Lamp1 (eBioscience, 1D4B, 1/1000, IF): validated for mouse Lamp1 by WB. |

# Eukaryotic cell lines

Policy information about cell lines

| Cell line source(s) | HEK293T cells were from ATCC. Immortalized MEFs were described in (PMID: 27324217). Vero cells were described in (PMID: 32994400). Raw264.7 were from InvivoGen. MRC-5 cells were from the Riken BioResource Center. |
|---|---|
| Authentication | Authentication of HEK293T cells was performed by ATCC with the short tandem repeat profiling. MEFs were identified by genotyping. Authentication of MRC-5 cells was performed by the Riken BioResource Center with the short tandem repeat profiling. No methods was used for authentication for Raw264.7. |
| Mycoplasma contamination | Confirm that all cell lines were tested negative for mycoplasma contaminations. |
| Commonly misidentified lines (See ICLAC register) | No commonly misidentified cell lines were used in the study. |

# Animals and other organisms

Policy information about studies involving animals; ARRIVE guidelines recommended for reporting animal research

| Laboratory animals | C57BL/6 mice |
|---|---|
| Wild animals | This study does not involve wild animals. |
| Field-collected samples | This study does not involve field-collected samples. |
| Ethics oversight | Ethics number PA17-84 approved by the Institute of Medical Sciences of the University of Tokyo. |

Note that full information on the approval of the study protocol must also be provided in the manuscript.

# Human research participants

Policy information about studies involving human research participants

| Population characteristics | Healthy human samples were collected from four Japanese males aged 30 - 55 with no significant medical history. |
|---|---|
| Recruitment | Participants were recruited from individuals working at Pediatric department of Kyoto University Hospital by word of mouth. |
| Ethics oversight | All experiments involving human subjects were conducted in accordance with the principles of the Declaration of Helsinki and were approved by the ethics committee of Kyoto University Hospital (protocol number: G1233). Written informed consent was obtained from the participants before sampling. No compensation was provided. |

Note that full information on the approval of the study protocol must also be provided in the manuscript.

# Flow Cytometry

## Plots

Confirm that:

☐ The axis labels state the marker and fluorochrome used (e.g. CD4-FITC).

☐ The axis scales are clearly visible. Include numbers along axes only for bottom left plot of group (a 'group' is an analysis of identical markers).

☐ All plots are contour plots with outliers or pseudocolor plots.

☐ A numerical value for number of cells or percentage (with statistics) is provided.

## Methodology

| | |
|---|---|
| Sample preparation | Sting-/- MEFs reconstituted with mRuby3-STING were treated with indicated siRNA for 54 h followed by stimulation with or without DMXAA for 18 h. Cells were detached with trypsin/EDTA and fixed with 4% PFA in PBS at room temperature for 15 min. |
| Instrument | Cell Sorter SH800 (Sony) |
| Software | Software version 2.1(Cell Sorter SH800 (Sony)) |
| Cell population abundance | *Describe the abundance of the relevant cell populations within post-sort fractions, providing details on the purity of the samples and how it was determined.* |
| Gating strategy | No gating for FSC/SSC. No gating for the signal of mRuby3. |

☐ Tick this box to confirm that a figure exemplifying the gating strategy is provided in the Supplementary Information.

