## [Peer Review File · Nature Cell Biology]

Peer Review Information

Journal: Nature Cell Biology

Manuscript Title: STING signalling is terminated through ESCRT-dependent microautophagy of vesicles originating from recycling endosomes

Corresponding author name(s): Dr Tomohiko Taguchi

Editorial Notes:

Redactions – transferred manuscripts (mention of previous referee reports from elsewhere) This manuscript has been previously reviewed at another journal. This document only contains reviewer comments, rebuttal and decision letters for versions considered at *Nature Cell Biology*. Mentions of prior referee reports have been redacted

Reviewer Comments & Decisions:

Decision Letter, initial version:

Dear Dr Taguchi,

Thank you for your patience as we were waiting for input from new reviewers on your manuscript "Lysosomal vesiculophagy terminates STING signalling". It has now been seen by 2 referees, who are experts in STING/autophagy (Referee #1); lysosomes/microautophagy (Referee #2), and whose comments are pasted below. (Please note that, as discussed previously, despite the numbering as Revs#1/#2 in our manuscript system, these are experts newly recruited to the panel -- these are not the "Reviewer #1" and "Reviewer #2" from the panel of experts who reviewed the work at [Redacted].) In light of their advice, we regret that we cannot offer to publish the study in Nature Cell Biology.

As you will see, the reviewers found the work interesting and felt that some of the points from the referees at [Redacted] were adequately addressed, and Rev#2 was very positive. However, Reviewer #1 was not convinced that the data sufficiently strongly supported the model. In particular, this reviewer felt that [redacted] concern that the vesicles could be derived via MVB formation was not sufficiently excluded. We have discussed the referee feedback in detail editorially and find that Reviewer #1's reservations about the strength of the evidence for the novel claims, echoing [redacted] previous concern, are regrettably too significant and preclude publication in NCB.

Although we regrettably cannot offer to publish your paper in *Nature Cell Biology*, you might wish to consider our sister journal *Nature Communications* as a potential venue for the publication of these results. *Nature Communications* publishes high quality and influential research across the full spectrum of the natural sciences. More information on the journal, the potential benefits of transfer and a link to transfer your paper, can be found at the bottom of this email. Please note that the editorial team at *Nature Communications* will consider your manuscript independently of our suggestion to transfer.

Please let me know if you would like me to consult with my colleagues at *Nature Communications* (or another sister journal) about a potential transfer of the study.

We are very sorry that we could not be more positive on this occasion, but we thank you for the opportunity to consider this work.

With kind regards,
Melina

Melina Casadio, PhD

Senior Editor, Nature Cell Biology

ORCID ID: <https://orcid.org/0000-0003-2389-2243>

Reviewers' comments

Reviewer #1 (Remarks to the Author):

In this manuscript Kochitsu, Y., et al., provide data on STING lysosomal-mediated degradation, which corroborates the previous findings reported by Gonugunta, V.K., et al., 2017. The authors also present novel evidence that suggests STING degradation is hindered by the absence of certain members of the ESCRT machinery. However, the manuscript shows only weak evidence of the proposed novel model that STING vesicles can be directly engulfed by lysosomes. Even though some images and videos consistent with this idea can be observed in the manuscript, they do not present any clear quantitative evidence of direct lysosome engulfment - these vesicles could be derived via multivesicular body formation. This same point is made by [redacted] and is not addressed by the authors in the latest rebuttal letter. Previous work in animal cells suggests that cellular material (Sahu, R., et al. (2011). Doi: 10.1016/j.devcel.2010.12.003, Mejlvang, J. et al. (2018). Doi: 10.1083/jcb.201711002) can be delivered to late endosomes during MVB formation and that this process requires the ESCRT machinery, similar to what is shown by Kochitsu, Y., et al. in this manuscript. However, the authors do not test if STING vesicles were generated as MVBs in Late Endosomal compartments. The authors propose a mechanism in which the Recycling Endosome is required, but do not provide evidence nor the mechanism by which the RE is required for lysosomes to engulf STING vesicles.

Minor point

STING ubiquitination in K288 has been described to be required for STING degradation but the reference to the prior work is missing (Li, Q., et al. (2018). Doi: 10.1038/s41421-018-0010-9).

Reviewer #2 (Remarks to the Author):

How to turn off STING has fascinated cell biologists and immunologists, including this reviewer, for a long time. We know it is degraded by lysosomes, but the detailed mechanism remains unclear. This manuscript by Kochitsu and co-workers represents a big step forward. It is through "lysosomal vesiculophagy," a cluster of recycling endosome-derived vesicles that are engulfed by lysosomal microautophagy. The authors demonstrated that ESCRT machinery is critical for this process, and the protein ubiquitination on lysine 288 of STING is also critical for the degradation. The data presented here are convincing and of high quality. Therefore, I am in favor of publishing this article in NCB.

I also looked into the review article mentioned by [redacted] to assess the novelty issue. In short, I agree with the authors in their rebuttal response. Fig. 5 in the review written by Hopfner and Hornung described activated STING could stimulate ATG5-dependent macroautophagy, which is a different question. What this paper discovered is the mechanism to degrade STING. And it is through an ATG5-independent but ESCRT-dependent microautophagy. In my opinion, the review article suggested by [redacted] only confirms the novelty of this study. [Redacted]'s comment that DMXAA might cause artifacts is a valid concern. But I think the authors have sufficiently addressed that critique.

Lastly, the weakness of the paper is the exact role of the ESCRT machinery during STING degradation. In particular, is ESCRT machinery driving the vesiculophagy process by invaginating the lysosomal membrane, or is it only involved in the final closure step of the microautophagy?

***Nature Communications* is the Nature Portfolio flagship Open Access journal. If you would like this work to be considered for publication there, you can easily transfer the manuscript by following the instructions below. It is not necessary to reformat your paper. Once all files are received, the editors at *Nature Communications* will assess your manuscript's suitability for potential publication; they aim to provide feedback quickly, with a median decision time of 8 days for first editorial decisions on suitability. The Nature Cell Biology referee reports will also be transferred and assessed by the editorial team. In some cases, papers are accepted without further peer review, providing a rapid path to publication. The journal offers double blind and transparent peer review options. For information on journal metrics please visit our Nature journals metrics page). Our open access pages contain information about article processing charges, open access funding, and advice and support from Springer Nature.

**I suggest that you consider Nature Communications as a suitable venue for your work. To transfer your manuscript there, please use our manuscript transfer portal. You will not have to re-supply manuscript metadata and files, unless you wish to make modifications, but please note that this link can only be used once and remains active until used. For more information, please see our manuscript transfer FAQ page.

Note that any decision to opt in to In Review at the original journal is not sent to the receiving journal on transfer. You can opt in to In Review at receiving journals that support this service by choosing to modify your manuscript on transfer. In Review is available for primary research manuscript types only.

**For Nature Portfolio general information and news for authors, see <http://npg.nature.com/authors>.

Author Rebuttal to Initial comments

Reviewers' comments

Reviewer #1 (Remarks to the Author):

In this manuscript Kochitsu, Y., et al., provide data on STING lysosomal-mediated degradation, which corroborates the previous findings reported by Gonugunta, V.K., et al., 2017. The authors also present novel evidence that suggests STING degradation is hindered by the absence of certain members of the ESCRT machinery. However, the manuscript shows only weak evidence of the proposed novel model that STING vesicles can be directly engulfed by lysosomes. Even though some images and videos consistent with this idea can be observed in the manuscript, they do not present any clear quantitative evidence of direct lysosome engulfment - these vesicles could be derived via multivesicular body formation. This same point is made by [redacted] and is not addressed by the authors in the latest rebuttal letter. Previous work in animal cells suggests that cellular material (Sahu, R., et al. (2011). Doi: 10.1016/j.devcel.2010.12.003, Mejlvang, J. et al. (2018). Doi: 10.1083/jcb.201711002) can be delivered to late endosomes during MVB formation and that this process requires the ESCRT machinery, similar to what is shown by Kochitsu, Y., et al. in this manuscript. However, the authors do not test if STING vesicles were generated as MVBs in Late Endosomal compartments.

> Thank you for the critical comments. The issue whether the MVB pathway, instead of lysosomal microautophagy as we propose, may be the main degradation pathway of STING, had also been raised by the previous reviewers by [redacted]. We appreciate that this is the most critical issue that has not been solved in the previous manuscript.

To address this question, we extensively examined if STING was co-localized with early endosomes and/or late endosomes at any time point up to 12 h after stimulation. We exploited Rab5 (the authentic early endosomal protein), EEA1 (an Rab5-effector localized primarily at early endosomes) and LBPA (the phospholipid preferentially localized at late endosomes) to visualize "endosomes". As shown in the new data in Figure 1h and Extended Data Fig. 1n, together with the data in Figure 1i (Extended Data Fig. 3b in the previous manuscript) showed that the association of STING with "endosomes" was minimum throughout the STING trafficking from the ER to lysosomes. Thus, these results suggested that the MVB pathway was not the principal mechanism of STING degradation. We revised the main text as follows:

Page 5 line 1J:

"We then examined whether lysosomes and/or endosomes encapsulated STING. Cells were stably transduced with mRuby3-STING, mTagBFP2-Rab5, and Lamp1-EGFP, so that endosomes and lysosomes were simultaneously monitored. Three hours after DMXAA stimulation, when STING started to be in acidic compartments (Fig. 1e), STING was found inside Lamp1-positive lysosomes, but not inside Rab5-positive endosomes (Fig. 1h). The quantitation also revealed that at any time point up to 12 h after stimulation, STING was not found inside Rab5-positive early endosomes (Fig. 1h-i, Extended Data Fig. 3), EEA1-positive early endosomes (Extended Data Fig. 1n), or LBPA-positive late endosomes (Extended Data Fig. 1n). These results suggested that lysosomes, not endosomes, directly encapsulated STING for degradation."

The negligible contribution of the MVB pathway to STING degradation is also supported by the experiments with a PI3K inhibitor wortmannin, which inhibits the invagination of early endosomal membrane (Futter *et al.*, *J Cell Biol* 2001, PMID 11756475; Fernandez-Borja *et al.*, *Curr Biol* 1999, PMID 9889123; Katzmann *et al.*, *Nat Rev Mol Cell Biol* 2002, PMID 12461556), generating intraluminal vesicles (Extended Data Fig. 1i-m).

Thus, we could mostly rule out the contribution of the three pathways (macroautophagy, membrane fusion, and the MVB pathway) that potentially operated to transfer STING in the cytoplasm to the lysosomal lumen. We believe that the present manuscript, with the additional sets of the experiments to examine the MVB pathway, firmly demonstrates that lysosomal microautophagy is the principal mechanism of STING degradation.

The authors propose a mechanism in which the Recycling Endosome is required, but do not provide evidence nor the mechanism by which the RE is required for lysosomes to engulf STING vesicles.

>Thank you for the critical comment. This is a very interesting question and we also would like to solve this.

Given that the size of the STING vesicles was around 100 nm and that they had electron-dense coats (Figure 2), we reasoned that they may be clathrin-coated vesicles. And indeed, we have a strong evidence to support this hypothesis.

STING co-localized very well with endogenous clathrin, 2 h after stimulation (a). Knockdown of clathrin heavy chain (HC) impaired STING degradation, as knockdown of Tsg101 (b). Perhaps, most importantly, in cells depleted of clathrin-HC, STING accumulated together with TfnR (c). These results indicated that (i) STING vesicles were clathrin-coated vesicles, and (ii) they were generated from RE membranes. In other words, RE functions to make STING vesicles, destined for degradation by lysosomes.

a

b

c

To characterize how RE is involved in STING degradation from the point of the STING vesiculation is critical for the advance of this study, however, we feel that it is a bit out of focus in the present manuscript. We would like to reserve these data for the future work.

Minor point

STING ubiquitination in K288 has been described to be required for STING degradation but the reference to the prior work is missing (Li, Q., et al. (2018). Doi: 10.1038/s41421-018-0010-9).

> Thank you for the comment. We included the publication by Li *et al.* as follows.

Page 9 line 8J:

"Among them, K288R mutant entirely lost the stimulation-dependent ubiquitination (Fig. 5d), and most importantly, was resistant to degradation (Fig. 5d-e), in line with the previous report with HEK293T cells³²."

Reviewer #2 (Remarks to the Author):

How to turn off STING has fascinated cell biologists and immunologists, including this reviewer, for a long time. We know it is degraded by lysosomes, but the detailed mechanism remains unclear. This manuscript by Kuchitsu and co-workers represents a big step forward. It is through "lysosomal vesiculophagy," a cluster of recycling endosome-derived vesicles that are engulfed by lysosomal microautophagy. The authors demonstrated that ESCRT machinery is critical for this process, and the protein ubiquitination on lysine 288 of STING is also critical for the degradation. The data presented here are convincing and of high quality. Therefore, I am in favor of publishing this article in NCB.

I also looked into the review article mentioned by [Redacted] to assess the novelty issue. In short, I agree with the authors in their rebuttal response. Fig. 5 in the review written by Hopfner and Hornung described activated STING could stimulate ATG5-dependent macroautophagy, which is a different question. What this paper discovered is the mechanism to degrade STING. And it is through an ATG5-independent but ESCRT-dependent microautophagy. In my opinion, the review article suggested by [redacted] only confirms the novelty of this study. [Redacted]'s comment that DMXAA might cause artifacts is a valid concern. But I think the authors have sufficiently addressed that critique.

Lastly, the weakness of the paper is the exact role of the ESCRT machinery during STING degradation. In particular, is ESCRT machinery driving the vesiculophagy process by invaginating the lysosomal membrane, or is it only involved in the final closure step of the microautophagy?

> Thank you for the very positive comments.

As Reviewer #2 pointed out, the exact site of action of the ESCRT machinery remains to be elucidated. We would like to solve this critical issue by visualizing several subunits of the ESCRT complex during lysosomal microautophagy. This experiment would also help us understand how the ESCRT complex can function on the lysosomal membrane, not on other membranes such as early/late endosomes, during STING degradation. Thank you very much again for the critical suggestion.

Decision Letter, first revision:

Dear Dr Taguchi,

Thank you for your email asking us to reconsider our decision on your manuscript, "Lysosomal vesiculophagy terminates STING signalling". We are always willing to hear the authors' perspective, but we must first prioritize decisions on new submissions. We appreciate your patience while we considered this appeal.

I have now discussed your manuscript, and the referees' comments and your rebuttal, in detail with my colleagues, and we would be willing to reconsider a revised manuscript provided that nothing similar is accepted for publication at Nature Cell Biology or published elsewhere in the meantime.

Our typical policy is to have all papers limited to 1 round of major experimental revision. Given the history of the manuscript, please note that we will look for strong reviewer support to move forward for publication. In other words, the referees will need to be fully convinced of the strength of the conclusions and quality of the data without the need for additional major revision.

In addition, please pay close attention to our guidelines on statistical and methodological reporting (listed below) as failure to do so may delay the reconsideration of the revised manuscript. In particular, please provide:

- a Supplementary Figure including unprocessed images of all gels/blots in the form of a multi-page pdf file. Please ensure that blots/gels are labeled and the sections presented in the figures are clearly indicated.
- a Supplementary Table including all numerical source data in Excel format, with data for different figures provided as different sheets within a single Excel file. The file should include source data giving rise to graphical representations and statistical descriptions in the paper and for all instances where the figures present representative experiments of multiple independent repeats, the source data of all repeats should be provided.

On resubmission please provide the completed Editorial Policy Checklist (found here <https://www.nature.com/documents/nr-editorial-policy-checklist.pdf>), and Reporting Summary (found here <https://www.nature.com/documents/nr-reporting-summary.pdf>). This is essential for reconsideration of the manuscript and these documents will be available to editors and referees in the event of peer review. For more information see below. Please also ensure that the presentation of statistical information in the revised submission complies with Nature Cell Biology's statistical guidelines (see below).

Please use the link below to submit the complete manuscript files, [IF RELEVANT: and include a point-by-point response to the complete reviewer comments, verbatim as provided in their reports.

[Redacted]

Please let us know how you wish to proceed and when we can expect your revised manuscript.

With kind regards,

Melina

Melina Casadio, PhD
Senior Editor, Nature Cell Biology
ORCID ID: <https://orcid.org/0000-0003-2389-2243>

GUIDELINES FOR EXPERIMENTAL AND STATISTICAL REPORTING

REPORTING REQUIREMENTS – To improve the quality of methods and statistics reporting in our papers we have recently revised the reporting checklist we introduced in 2013. We are now asking all life sciences authors to complete two items: an Editorial Policy Checklist (found here <https://www.nature.com/documents/nr-editorial-policy-checklist.pdf>) that verifies compliance with all required editorial policies and a reporting summary (found here <https://www.nature.com/documents/nr-reporting-summary.pdf>) that collects information on experimental design and reagents. These documents are available to referees to aid the evaluation of the manuscript. Please note that these forms are dynamic 'smart pdfs' and must therefore be downloaded and completed in Adobe Reader. We will then flatten them for ease of use by the reviewers. If you would like to reference the guidance text as you complete the template, please access these flattened versions at <http://www.nature.com/authors/policies/availability.html>.

Author Rebuttal, first revision:

Reviewers' comments

Reviewer #1 (Remarks to the Author):

In this manuscript Kochitsu, Y., et al., provide data on STING lysosomal-mediated degradation, which corroborates the previous findings reported by Gonugunta, V.K., et al., 2017. The authors also present novel evidence that suggests STING degradation is hindered by the absence of certain members of the ESCRT machinery. However, the manuscript shows only weak evidence of the proposed novel model that STING vesicles can be directly engulfed by lysosomes. Even though some images and videos consistent with this idea can be observed in the manuscript, they do not present any clear quantitative evidence of direct lysosome engulfment - these vesicles could be derived via multivesicular body formation. This same point is made by [redacted] and is not addressed by the authors in the latest rebuttal letter. Previous work in animal cells suggests that cellular material (Sahu, R., et al. (2011). Doi: 10.1016/j.devcel.2010.12.003, Mejlvang, J. et al. (2018). Doi: 10.1083/jcb.201711002) can be delivered to late endosomes during MVB formation and that this process requires the ESCRT machinery, similar to what is shown by Kochitsu, Y., et al. in this manuscript. However, the authors do not test if STING vesicles were generated as MVBs in Late Endosomal compartments.

> Thank you for the critical comments. The issue whether the MVB pathway, instead of lysosomal microautophagy as we propose, may be the main degradation pathway of STING, had also been raised by the previous reviewers by [redacted]. We appreciate that this is the most critical issue that has not been solved in the previous manuscript.

To address this question, we extensively examined if STING was co-localized with early endosomes and/or late endosomes at any time point up to 12 h after stimulation. We exploited Rab5 (the authentic early endosomal protein), EEA1 (an Rab5-effector localized primarily at early endosomes) and LBPA (the phospholipid preferentially localized at late endosomes) to visualize "endosomes". As shown in the new data in Figure 1h and Extended Data Fig. 1n, together with the data in Figure 1i (Extended Data Fig. 3b in the previous manuscript) showed that the association of STING with "endosomes" was minimum throughout the STING trafficking from the ER to lysosomes. Thus, these results suggested that the MVB pathway was not the principal mechanism of STING degradation. We revised the main text as follows:

Page 5 line 1J:

"We then examined whether lysosomes and/or endosomes encapsulated STING. Cells were stably transduced with mRuby3-STING, mTagBFP2-Rab5, and Lamp1-EGFP, so that endosomes and lysosomes were simultaneously monitored. Three hours after DMXAA stimulation, when STING started to be in acidic compartments (Fig. 1e), STING was found inside Lamp1-positive lysosomes, but not inside Rab5-positive endosomes (Fig. 1h). The quantitation also revealed that at any time point up to 12 h after stimulation, STING was not found inside Rab5-positive early endosomes (Fig. 1h-i, Extended Data Fig. 3), EEA1-positive early endosomes (Extended Data Fig. 1n), or LBPA-positive late endosomes (Extended Data Fig. 1n). These results suggested that lysosomes, not endosomes, directly encapsulated STING for degradation."

The negligible contribution of the MVB pathway to STING degradation is also supported by the experiments with a PI3K inhibitor wortmannin, which inhibits the invagination of early endosomal membrane (Futter *et al.*, *J Cell Biol* 2001, PMID 11756475; Fernandez-Borja *et al.*, *Curr Biol* 1999, PMID 9889123; Katzmann *et al.*, *Nat Rev Mol Cell Biol* 2002, PMID 12461556), generating intraluminal vesicles (Extended Data Fig. 1i-m).

Thus, we could mostly rule out the contribution of the three pathways (macroautophagy, membrane fusion, and the MVB pathway) that potentially operated to transfer STING in the cytoplasm to the lysosomal lumen. We believe that the present manuscript, with the additional sets of the experiments to examine the MVB pathway, firmly demonstrates that lysosomal microautophagy is the principal mechanism of STING degradation.

The authors propose a mechanism in which the Recycling Endosome is required, but do not provide evidence nor the mechanism by which the RE is required for lysosomes to engulf STING vesicles.

>Thank you for the critical comment. This is a very interesting question and we also would like to solve this.

Given that the size of the STING vesicles was around 100 nm and that they had electron-dense coats (Figure 2), we reasoned that they may be clathrin-coated vesicles. And indeed, we have a strong evidence to support this hypothesis.

STING co-localized very well with endogenous clathrin, 2 h after stimulation (a). Knockdown of clathrin heavy chain (HC) impaired STING degradation, as knockdown of Tsg101 (b). Perhaps, most importantly, in cells depleted of clathrin-HC, STING accumulated together with TfnR (c). These results indicated that (i) STING vesicles were clathrin-coated vesicles, and (ii) they were generated from RE membranes. In other words, RE functions to make STING vesicles, destined for degradation by lysosomes.

a

b

c

To characterize how RE is involved in STING degradation from the point of the STING vesiculation is critical for the advance of this study, however, we feel that it is a bit out of focus in the present manuscript. We would like to reserve these data for the future work.

Minor point

STING ubiquitination in K288 has been described to be required for STING degradation but the reference to the prior work is missing (Li, Q., et al. (2018). Doi: 10.1038/s41421-018-0010-9).

> Thank you for the comment. We included the publication by Li *et al.* as follows.

Page 9 line 8J:

"Among them, K288R mutant entirely lost the stimulation-dependent ubiquitination (Fig. 5d), and most importantly, was resistant to degradation (Fig. 5d-e), in line with the previous report with HEK293T cells³²."

Reviewer #2 (Remarks to the Author):

How to turn off STING has fascinated cell biologists and immunologists, including this reviewer, for a long time. We know it is degraded by lysosomes, but the detailed mechanism remains unclear. This manuscript by Kuchitsu and co-workers represents a big step forward. It is through "lysosomal vesiculophagy," a cluster of recycling endosome-derived vesicles that are engulfed by lysosomal microautophagy. The authors demonstrated that ESCRT machinery is critical for this process, and the protein ubiquitination on lysine 288 of STING is also critical for the degradation. The data presented here are convincing and of high quality. Therefore, I am in favor of publishing this article in NCB.

I also looked into the review article mentioned by [redacted] to assess the novelty issue. In short, I agree with the authors in their rebuttal response. Fig. 5 in the review written by Hopfner and Hornung described activated STING could stimulate ATG5-dependent macroautophagy, which is a different question. What this paper discovered is the mechanism to degrade STING. And it is through an ATG5-independent but ESCRT-dependent microautophagy. In my opinion, the review article suggested by [redacted] only confirms the novelty of this study. [Redacted]'s comment that DMXAA might cause artifacts is a valid concern. But I think the authors have sufficiently addressed that critique.

Lastly, the weakness of the paper is the exact role of the ESCRT machinery during STING degradation. In particular, is ESCRT machinery driving the vesiculophagy process by invaginating the lysosomal membrane, or is it only involved in the final closure step of the microautophagy?

> Thank you for the very positive comments.

As Reviewer #2 pointed out, the exact site of action of the ESCRT machinery remains to be elucidated. We would like to solve this critical issue by visualizing several subunits of the ESCRT complex during lysosomal microautophagy. This experiment would also help us understand how the ESCRT complex can function on the lysosomal membrane, not on other membranes such as early/late endosomes, during STING degradation. Thank you very much again for the critical suggestion.

Decision Letter, second revision:

Dear Dr Taguchi,

Thank you for submitting your revised manuscript, "Lysosomal vesiculophagy terminates STING signalling", to the journal and thank you for your patience with the process. The manuscript has now been seen by our returning Reviewer #1 (i.e., the Reviewer #1 from peer review at NCB - not [Redacted]'s Rev#1). As you will see from their comments (attached below), they are supportive of the work, but are not yet convinced by the vesiculophagy model. We have discussed this reviewer's comments in depth, and we would like to invite you to submit a final revised version of the study with text edits to tone down the lysosome model. We feel it will be needed to rewrite the title, abstract, and manuscript text throughout to remove the conclusion about vesiculophagy and the lysosome model, to make sure the body of work is described accurately and consistently throughout (we otherwise worry that editing only the title as suggested by the rev would be confusing to readers). For a revised title, we suggest "STING signaling is terminated through ESCRT-dependent delivery of vesicles of recycling endosome origin" or "STING signaling is terminated through ESCRT-dependent delivery of vesicles originating from recycling endosomes", or a title along these lines. No further experimentation is required for publication in our view.

One additional change to the text we recommend is to please include text discussion of the recent work by Liu, Xu et al "Clathrin-associated AP-1 controls termination of STING signalling" (doi: 10.1038/s41586-022-05354-0). Perhaps integrating your data currently shown in the rebuttal in the manuscript (if the data are strong enough) would also make sense in that light - however, whether to share the data in the paper at this stage is entirely up to you.

Please let us know if you have any questions about the changes needed at this stage. When resubmitting the manuscript, please make sure to provide the reporting summary, editorial policy checklist, source blot images, and source numerical data as done for this resubmission and as explained below.

Please pay close attention to our guidelines on statistical and methodological reporting (listed below) as failure to do so may delay the reconsideration of the revised manuscript. In particular please provide:

We therefore invite you to take these points into account when revising the manuscript. In addition, when preparing the revision please:

- ensure that it conforms to our format instructions and publication policies (see below and www.nature.com/nature/authors/).

- provide a point-by-point rebuttal to the full referee reports verbatim, as provided at the end of this letter.

- provide the completed Editorial Policy Checklist (found here <https://www.nature.com/authors/policies/Policy.pdf>), and Reporting Summary (found here <https://www.nature.com/authors/policies/ReportingSummary.pdf>). This is essential for reconsideration of the manuscript and these documents will be available to editors and referees in the event of peer review. For more information see <http://www.nature.com/authors/policies/availability.html> or contact me.

Nature Cell Biology is committed to improving transparency in authorship. As part of our efforts in this direction, we are now requesting that all authors identified as 'corresponding author' on published papers create and link their Open Researcher and Contributor Identifier (ORCID) with their account on the Manuscript Tracking System (MTS), prior to acceptance. ORCID helps the scientific community achieve unambiguous attribution of all scholarly contributions. You can create and link your ORCID from the home page of the MTS by clicking on 'Modify my Springer Nature account'. For more information please visit www.springernature.com/orcid.

[redacted]

We would like to receive the revision within four weeks. If submitted within this time period, reconsideration of the revised manuscript will not be affected by related studies published elsewhere, or accepted for publication in Nature Cell Biology in the meantime. We would be happy to consider a revision even after this timeframe, but in that case we will consider the published literature at the time of resubmission when assessing the file.

We hope that you will find our referees' comments and editorial guidance helpful. Please do not hesitate to contact me if there is anything you would like to discuss. Thank you again for considering the journal for your work.

Best wishes,

Melina

Melina Casadio, PhD
Senior Editor, Nature Cell Biology
ORCID ID: <https://orcid.org/0000-0003-2389-2243>

Reviewers' Comments:

Reviewer #1:

Remarks to the Author:

The authors have addressed some of my comments in their revision. I still think the direct vesiculophagy to lysosome aspect is fairly weak. If the authors left the results and abstract as is but changed the title to focus on the ESCRT machinery and delete vesiculophagy and lysosome model, I would support acceptance.

GUIDELINES FOR SUBMISSION OF NATURE CELL BIOLOGY ARTICLES

ARTICLE FORMAT

ABSTRACT – should not exceed 150 words and should be unreferenced. This paragraph is the most visible part of the paper and should briefly outline the background and rationale for the work, and accurately summarize the main results and conclusions. Key genes, proteins and organisms should be specified to ensure discoverability of the paper in online searches.

TEXT – the main text consists of the Introduction, Results, and Discussion sections and must not exceed 3500 words including the abstract. The Introduction should expand on the background relating to the work. The Results should be divided in subsections with subheadings, and should provide a concise and accurate description of the experimental findings. The Discussion should expand on the findings and their implications. All relevant primary literature should be cited, in particular when discussing the background and specific findings.

REFERENCES – are limited to a total of 70 in the main text and Methods combined,. They must be numbered sequentially as they appear in the main text, tables and figure legends and Methods and must follow the precise style of Nature Cell Biology references. References only cited in the Methods should be numbered consecutively following the last reference cited in the main text. References only associated with Supplementary Information (e.g. in supplementary legends) do not count toward the total reference limit and do not need to be cited in numerical continuity with references in the main text. Only published papers can be cited, and each publication cited should be included in the numbered reference list, which should include the manuscript titles. Footnotes are not permitted.

Methods should be written concisely, but should contain all elements necessary to allow interpretation and replication of the results. As a guideline, Methods sections typically do not exceed 3,000 words. The Methods should be divided into subsections listing reagents and techniques. When citing previous methods, accurate references should be provided and any alterations should be noted. Information must be provided about: antibody dilutions, company names, catalogue numbers and clone numbers for monoclonal antibodies; sequences of RNAi and cDNA probes/primers or company names and catalogue numbers if reagents are commercial; cell line names, sources and information on cell line identity and authentication. Animal studies and experiments involving human subjects must be reported in detail, identifying the committees approving the protocols. For studies involving human subjects/samples, a statement must be included confirming that informed consent was obtained. Statistical analyses and information on the reproducibility of experimental results should be provided in a section titled "Statistics and Reproducibility".

All Nature Cell Biology manuscripts submitted on or after March 21 2016, must include a Data availability statement as a separate section after Methods but before references, under the heading "Data Availability". For Springer Nature policies on data availability see <http://www.nature.com/authors/policies/availability.html>; for more information on this particular policy see <http://www.nature.com/authors/policies/data/data-availability-statements-data-citations.pdf>. The Data availability statement should include:

- Accession codes for primary datasets (generated during the study under consideration and designated as "primary accessions") and secondary datasets (published datasets reanalysed during the study under consideration, designated as "referenced accessions"). For primary accessions data should be made public to coincide with publication of the manuscript. A list of data types for which submission to community-endorsed public repositories is mandated (including sequence, structure, microarray, deep sequencing data) can be found here <http://www.nature.com/authors/policies/availability.html#data>.
- Unique identifiers (accession codes, DOIs or other unique persistent identifier) and hyperlinks for datasets deposited in an approved repository, but for which data deposition is not mandated (see here for details <http://www.nature.com/sdata/data-policies/repositories>).
- At a minimum, please include a statement confirming that all relevant data are available from the authors, and/or are included with the manuscript (e.g. as source data or supplementary information), listing which data are included (e.g. by figure panels and data types) and mentioning any restrictions on availability.
- If a dataset has a Digital Object Identifier (DOI) as its unique identifier, we strongly encourage including this in the Reference list and citing the dataset in the Methods.

We recommend that you upload the step-by-step protocols used in this manuscript to the Protocol Exchange. More details can found at www.nature.com/protocolexchange/about.

DISPLAY ITEMS – main display items are limited to 6-8 main figures and/or main tables. For Supplementary Information see below.

FIGURES – Colour figure publication costs \$395 per colour figure. All panels of a multi-panel figure must be logically connected and arranged as they would appear in the final version. Unnecessary figures and figure panels should be avoided (e.g. data presented in small tables could be stated briefly in the text instead).

All imaging data should be accompanied by scale bars, which should be defined in the legend. Cropped images of gels/blots are acceptable, but need to be accompanied by size markers, and to retain visible background signal within the linear range (i.e. should not be saturated). The boundaries of panels with low background have to be demarked with black lines. Splicing of panels should only be considered if unavoidable, and must be clearly marked on the figure, and noted in the legend with a statement on whether the samples were obtained and processed simultaneously. Quantitative comparisons between samples on different gels/blots are discouraged; if this is unavoidable, it has to be performed for samples derived from the same experiment with gels/blots were processed in parallel, which needs to be stated in the legend.

Regardless of format, all figures must be vector graphic compatible files, not supplied in a flattened raster/bitmap graphics format, but should be fully editable, allowing us to highlight/copy/paste all text and move individual parts of the figures (i.e. arrows, lines, x and y axes, graphs, tick marks, scale bars etc). The only parts of the figure that should be in pixel raster/bitmap format are photographic images or 3D rendered graphics/complex technical illustrations.

All placed images (i.e. a photo incorporated into a figure) should be on a separate layer and independent

from any superimposed scale bars or text. Individual photographic images must be a minimum of 300+ DPI (at actual size) or kept constant from the original picture acquisition and not decreased in resolution post image acquisition. All colour artwork should be RGB format.

Unprocessed scans of all key data generated through electrophoretic separation techniques need to be presented in a supplementary figure that should be labeled and numbered as the final supplementary figure, and should be mentioned in every relevant figure legend. This figure does not count towards the total number of figures and is the only figure that can be displayed over multiple pages, but should be provided as a single file, in PDF or TIFF format. Data in this figure can be displayed in a relatively informal style, but size markers and the figures panels corresponding to the presented data must be indicated.

The total number of Supplementary Figures (not including the “unprocessed scans” Supplementary Figure) should not exceed the number of main display items (figures and/or tables (see our Guide to Authors and March 2012 editorial <http://www.nature.com/ncb/authors/submit/index.html#suppinfo>; <http://www.nature.com/ncb/journal/v14/n3/index.html#ed>). No restrictions apply to Supplementary Tables or Videos, but we advise authors to be selective in including supplemental data.

GUIDELINES FOR EXPERIMENTAL AND STATISTICAL REPORTING

REPORTING REQUIREMENTS – To improve the quality of methods and statistics reporting in our papers we have recently revised the reporting checklist we introduced in 2013. We are now asking all life sciences authors to complete two items: an Editorial Policy Checklist (found here <https://www.nature.com/authors/policies/Policy.pdf>) that verifies compliance with all required editorial policies and a Reporting Summary (found here <https://www.nature.com/authors/policies/ReportingSummary.pdf>) that collects information on

experimental design and reagents. These documents are available to referees to aid the evaluation of the manuscript. Please note that these forms are dynamic 'smart pdfs' and must therefore be downloaded and completed in Adobe Reader. We will then flatten them for ease of use by the reviewers. If you would like to reference the guidance text as you complete the template, please access these flattened versions at <http://www.nature.com/authors/policies/availability.html>.

Author Rebuttal, second revision:

Reviewers' Comments:

Reviewer #1:

Remarks to the Author:

The authors have addressed some of my comments in their revision. I still think the direct vesiculophagy to lysosome aspect is fairly weak. If the authors left the results and abstract as is but changed the title to focus on the ESCRT machinery and delete vesiculophagy and lysosome model, I would support acceptance.

>Thank you very much for your comments. According to your suggestions, we revised the title to focus on the ESCRT machinery and deleted "lysosomal vesiculophagy model" throughout the text and Figures.

Decision Letter, third revision:

Our ref: NCB-A49159C

23rd November 2022

Dear Dr. Taguchi,

Thank you for submitting your revised manuscript "STING signalling is terminated through ESCRT-dependent microautophagy of vesicles originating from recycling endosomes" (NCB-A49159C). We have discussed the manuscript in depth again editorially and find the changes have improved and clarified the manuscript. We'll be happy in principle to publish it in Nature Cell Biology, pending minor revisions to comply with our editorial and formatting guidelines.

Please note that the current version of your manuscript is in a PDF format. Could you please email us a copy of the file in an editable format (Microsoft Word or LaTeX)? We cannot proceed with PDFs at this stage. Many thanks for your attention to this point.

Once we have the Word file, we will begin performing detailed checks on your paper and will send you a checklist detailing our editorial and formatting requirements in about 1-2 weeks. Please do not upload the final materials and make any revisions until you receive this additional information from us.

Thank you again for your interest in Nature Cell Biology. Please do not hesitate to contact me if you have any questions.

Sincerely,
Melina

Melina Casadio, PhD
Senior Editor, Nature Cell Biology
ORCID ID: <https://orcid.org/0000-0003-2389-2243>

Decision Letter, final checks:

Our ref: NCB-A49159C

2nd December 2022

Dear Dr. Taguchi,

Thank you for your patience as we've prepared the guidelines for final submission of your Nature Cell Biology manuscript, "STING signalling is terminated through ESCRT-dependent microautophagy of vesicles originating from recycling endosomes" (NCB-A49159C). Please carefully follow the step-by-

step instructions provided in the attached file, and add a response in each row of the table to indicate the changes that you have made. Please also check and comment on any additional marked-up edits we have proposed within the text. Ensuring that each point is addressed will help to ensure that your revised manuscript can be swiftly handed over to our production team.

In recognition of the time and expertise our reviewers provide to Nature Cell Biology's editorial process, we would like to formally acknowledge their contribution to the external peer review of your manuscript entitled "STING signalling is terminated through ESCRT-dependent microautophagy of vesicles originating from recycling endosomes". For those reviewers who give their assent, we will be publishing their names alongside the published article.

Nature Cell Biology offers a Transparent Peer Review option for new original research manuscripts submitted after December 1st, 2019. As part of this initiative, we encourage our authors to support increased transparency into the peer review process by agreeing to have the reviewer comments, author rebuttal letters, and editorial decision letters published as a Supplementary item. When you submit your final files please clearly state in your cover letter whether or not you would like to participate in this initiative. Please note that failure to state your preference will result in delays in accepting your manuscript for publication.

Cover suggestions

As you prepare your final files we encourage you to consider whether you have any images or illustrations that may be appropriate for use on the cover of Nature Cell Biology.

Nature Cell Biology has now transitioned to a unified Rights Collection system which will allow our Author Services team to quickly and easily collect the rights and permissions required to publish your work. Approximately 10 days after your paper is formally accepted, you will receive an email in providing you with a link to complete the grant of rights. If your paper is eligible for Open Access, our Author Services team will also be in touch regarding any additional information that may be required to arrange payment for your article.

Please note that *Nature Cell Biology* is a Transformative Journal (TJ). Authors may publish their research with us through the traditional subscription access route or make their paper immediately open access through payment of an article-processing charge (APC). Authors will not be required to make a final decision about access to their article until it has been accepted. Find out more about Transformative Journals

Please use the following link for uploading these materials:
[Redacted]

Best regards,

Kendra Donahue
Staff
Nature Cell Biology

On behalf of

Melina Casadio, PhD

Senior Editor, Nature Cell Biology
ORCID ID: <https://orcid.org/0000-0003-2389-2243>

Final Decision Letter:

Dear Dr Taguchi,

I am pleased to inform you that your manuscript, "STING signalling is terminated through ESCRT-dependent microautophagy of vesicles originating from recycling endosomes", has now been accepted for publication in Nature Cell Biology. Congratulations on this very nice work!

Please note that *Nature Cell Biology* is a Transformative Journal (TJ). Authors may publish their research with us through the traditional subscription access route or make their paper immediately open access through payment of an article-processing charge (APC). Authors will not be required to make a final decision about access to their article until it has been accepted. Find out more about

Transformative Journals

If you have not already done so, we strongly recommend that you upload the step-by-step protocols used in this manuscript to the Protocol Exchange (www.nature.com/protocolexchange), an open online resource established by Nature Protocols that allows researchers to share their detailed experimental know-how. All uploaded protocols are made freely available, assigned DOIs for ease of citation and are fully searchable through nature.com. Protocols and Nature Portfolio journal papers in which they are used can be linked to one another, and this link is clearly and prominently visible in the online versions of both papers. Authors who performed the specific experiments can act as primary authors for the Protocol as they will be best placed to share the methodology details, but the Corresponding Author of the present research paper should be included as one of the authors. By uploading your Protocols to Protocol Exchange, you are enabling researchers to more readily reproduce or adapt the methodology you use, as well as increasing the visibility of your protocols and papers. You can also establish a dedicated page to collect your lab Protocols. Further information can be found at www.nature.com/protocolexchange/about

With kind regards,

Melina

Melina Casadio, PhD
Senior Editor, Nature Cell Biology
ORCID ID: <https://orcid.org/0000-0003-2389-2243>

** Visit the Springer Nature Editorial and Publishing website at www.springernature.com/editorial-and-publishing-jobs for more information about our career opportunities. If you have any questions please click here.**